# Modeling cryo-EM structures in alternative states with AlphaFold2-based models and density-guided simulations
Tatiana Shugaeva [1], Rebecca J. Howard [1,2], Nandan Haloi [1] ✉ & Erik Lindahl [1,2] ✉

Modeling atomic coordinates into a target cryo-electron microscopy map is a crucial step in structure determination. Despite recent advances, proteins with multiple functional states remain a challenge - particularly when suitable molecular templates are unavailable for certain states, and the map resolution is not high enough to build de novo models. This is a common scenario, for example, among pharmacologically relevant membrane-bound receptors and transporters. Here, we introduce a refinement approach in which (i) several initial models are generated by stochastic subsampling of the multiple sequence alignment (MSA) space in AlphaFold2, (ii) the resulting models are subjected to structure-based k-means clustering, iii) density-guided molecular dynamics simulations are performed from the cluster representatives, and (iv) a final model is selected on the basis of both map fit and model quality. This results in improved fitting accuracy compared to single starting point scenarios for three membrane proteins (the calcitonin receptor-like receptor, L-type amino acid transporter and alanine-serine-cysteine transporter) which undergo substantial conformational transitions between functional states. Our results indicate that ensemble construction using generative AI combined with simulation-based refinement facilitates building of alternative states in several families of membrane proteins.

Cryo-electron microscopy (cryo-EM) is one of the most widely used and rapidly developing techniques for biomolecular structure determination. Advances in both hardware and algorithms have enabled density map reconstruction at or near atomistic resolution for numerous systems[1,2]. In parallel, automated as well as manual model building methods have provided detailed insights into molecular processes, complex architectures and new conformations[3]. Despite this progress, building structures remains challenging for systems where 1) a target protein undergoes conformational transitions between multiple states[4] and 2) the map resolution is not high enough for de novo model building[5–8].

Density-guided molecular dynamics (MD) simulations, where a biasing potential is added to the classical forcefield to move atoms toward the experimental map, might provide solutions to both aforementioned challenges[9–15]. However, the success of density-guided MD simulation depends on the reliability of the initial model. Even when a template structure resolved in one state is available, it may differ substantially from a target density resolved in another state, such that automated approaches produce poorly fitting or nonphysical models (Fig. 1—left). Such

scenarios may be common, for example, among membrane-bound receptor and transporter proteins, which constitute overrepresented but dynamically complex pharmaceutical targets[16]. Accuracy may be further improved by fitting a well-sampled ensemble rather than a single initial model (Fig. 1—right); however, the generation of such ensembles can be computationally expensive and time-consuming using classical MD or enhanced sampling simulations[17].

Here, we present an approach (Fig. 2) that combines generative artificial intelligence (AI) methods and flexible fitting to refine protein structures based on sequence information and low- to medium-resolution cryo-EM data. Taking advantage of recent developments in neural network-based structure prediction[18], we use stochastic subsampling of the multiple sequence alignment (MSA) depth in AlphaFold2[19] to generate a broader ensemble of potential starting structures, then run density-guided MD simulations from cluster representatives to select an optimized structure for a given experimental density. We demonstrate the applicability of this approach to three test cases, all membrane proteins that undergo conformational changes between experimentally resolved states. As reference,

[1]Department of Applied Physics, Science for Life Laboratory, KTH Royal Institute of Technology, Tomtebodavägen 23, Solna, SE-17165, Sweden. [2]Department of Biochemistry and Biophysics, Science for Life Laboratory, Stockholm University, Tomtebodavägen 23, Solna, SE-17165, Sweden.
✉e-mail: nandan.haloi@scilifelab.se; erik.lindahl@scilifelab.se

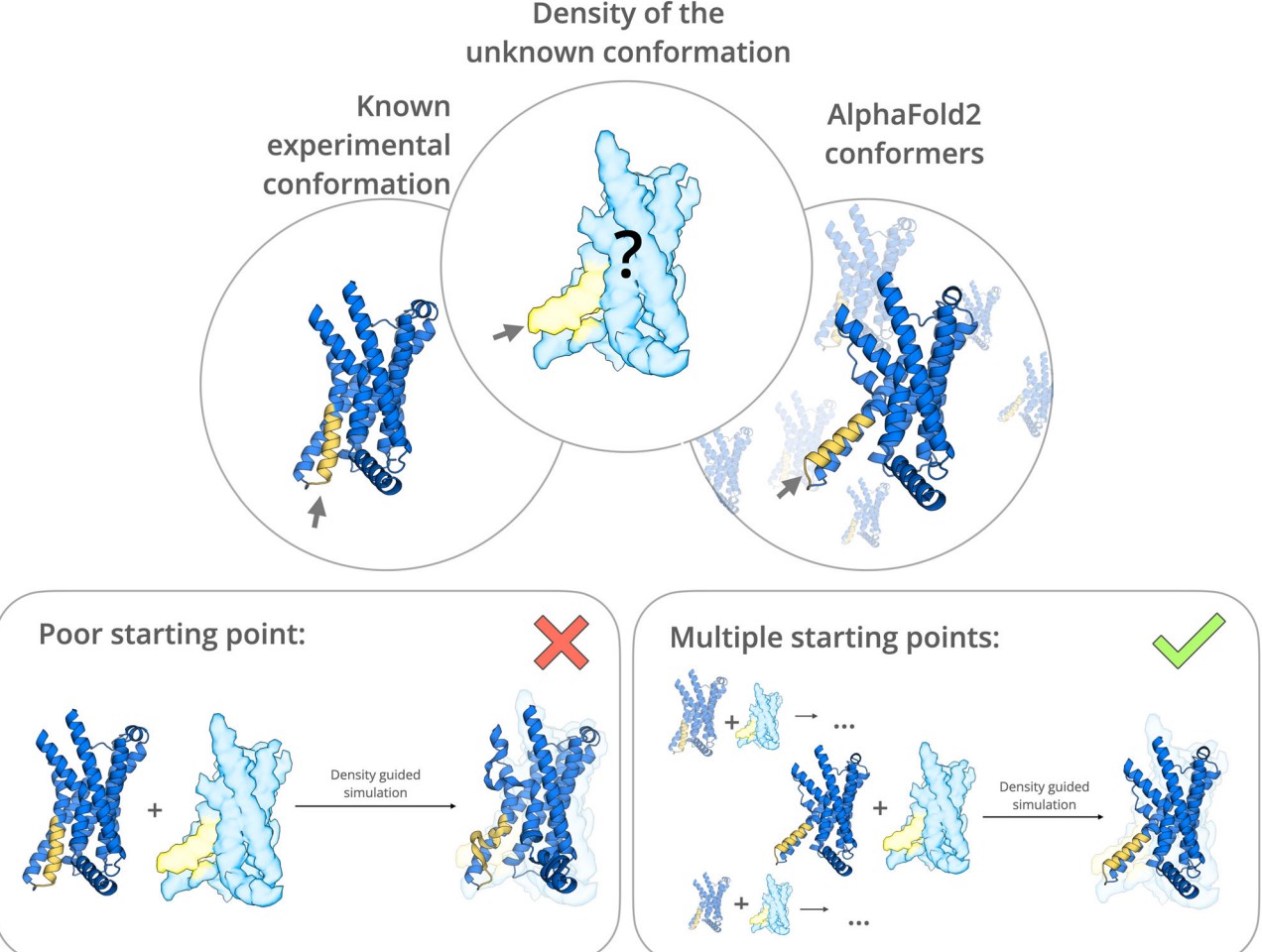

**Fig. 1 | Challenges and strategies in automated modeling of proteins occupying alternative conformations in cryo-EM maps.** *Top*, general representation of the common experimental problem in which a target cryo-EM density (center) is resolved for a protein with a known structure (left) that differs from the target in one or more regions, indicated by arrows; an AI-generated ensemble (right) may include models that more closely approximate the target. Bottom, detailed representation of a case in which density-guided simulations, represented by forward arrows, fail to fit a plausible model based on the known structure (left), but succeed on the basis of at least one member of the AI-generated ensemble (right).

default density-guided simulations in GROMACS failed to accurately fit the structure built in one state to the density resolved in another. In contrast, using our new approach, we successfully resolve state-dependent differences including the bending of an individual helix, rearrangement of neighboring helices, and reformation of an entire domain, demonstrating applicability to a range of systems and conformational dynamics.

## Results

### Incorporating generative models with flexible fitting

As a straightforward approach to incorporating AI-generated models with flexible fitting to determine challenging protein structures (Fig. 2—right), we first used stochastic subsampling of input MSA depth in AlphaFold2 to produce 1250 models for each of three protein test systems. To filter out substantially misfolded models, we prioritized those scoring better than (below) −100 on the basis of generalized orientation-dependent all-atom potential (GOAP) structure-quality scoring, as described in Methods. To identify a limited set of plausible models representative of the generated ensemble, we then aligned the filtered models to the structure in the already-known state, and clustered them by the $k$-means method on the basis of Cartesian coordinates[20]. Finally, we used the model closest to each cluster centroid (cluster representative) for fitting to an experimental density, termed the target state.

After rigid-body alignment to the relevant density, we subjected each representative model to density-guided MD simulations. Across each

simulation, we monitored the cross-correlation to the target map as a model-fitting metric, and selected the simulation with the highest mean cross-correlation for further analysis. The implementation of adaptive force scaling for fitting in GROMACS means the run will continue until the overall forces of the system are too large to be compatible with the integration time step, which can introduce distortions. To instead identify a balanced frame, we also monitor the GOAP score as a quality metric for structural geometry[21], as done in our previous study[22]. After normalizing both the fitting and geometry metrics to [0, 1], the sum of the normalized GOAP score and cross-correlation was computed as a compound score for each simulation frame, with higher scores representing a combination of good fit (high correlation) and geometry (minimal penalty due to overfitting). We selected the frame with the highest compound score as a final model.

To compare our approach to a more standard flexible fitting protocol[14], we performed parallel density-guided MD simulations for each experimental map using a single experimental structure of the same protein resolved under different conditions, termed the known state (Fig. 2—left). This scenario is common in investigation of challenging targets, in which a structure has been determined in one functional state, but deviates substantially from an experimental density representing a different state. As for many experimental structures, for two of our three test systems fitting required interpolation of loops not reported in the initial known model, a step rendered unnecessary in our generated ensemble approach described

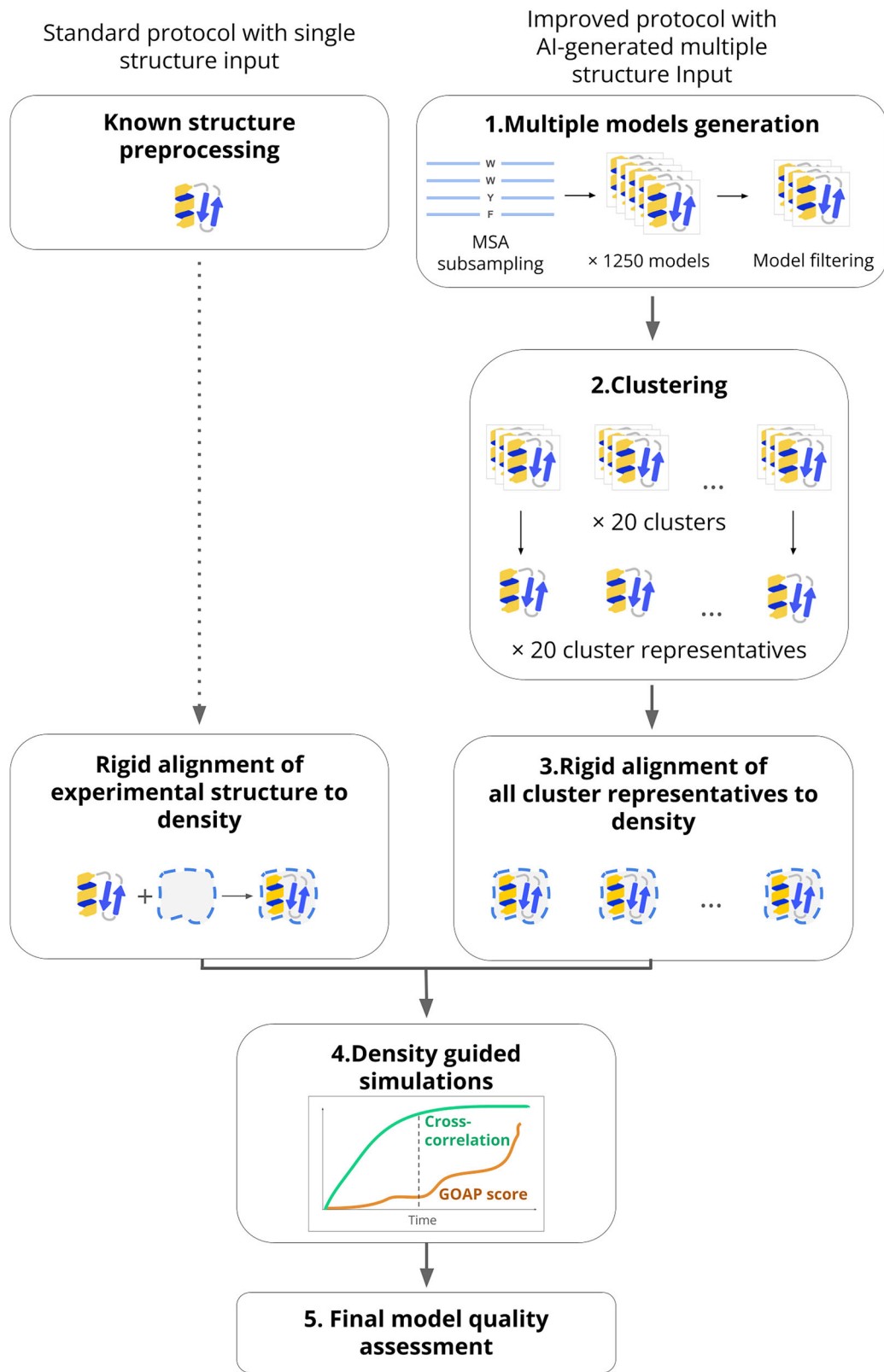

**Fig. 2 | Incorporating generative AI models with flexible fitting.** Left, in a standard approach to automated fitting of a new cryo-EM map, a known experimental structure is pre-processed, aligned to the target density, and subjected to density-guided MD simulations. Right, in the approach described here, an ensemble of models is generated by MSA subsampling in AlphaFold2, filtered using scaled GOAP score and clustered. Cluster representatives are aligned to the density and subjected to similar simulations. From the simulation with the highest mean cross-correlation to the target density, a final model is chosen to optimize map fit (cross-correlation) and model geometric quality (GOAP score). The protein emoji used in the scheme is taken from https://github.com/whitead/protein-emoji.

above. In both ensemble-based and single-model approaches, we refrained from applying secondary structure restraints. Such restraints are commonly used in density-guided simulations to prevent nonphysical structure disruptions[10,23], but can limit the accommodation of conformational transitions—for example, kinking of a helix, rearrangement among secondary structure elements, or remodeling of an entire domain.

We tested our approach by modeling publicly available experimental densities for three membrane proteins: the calcitonin receptor-like receptor (CLR, EMD 20906, 2.3 Å resolution)[24], L-type amino acid transporter (LAT1, EMD 30841, 3.4 Å resolution)[25] and alanine-serine-cysteine transporter 2 (ASCT2, EMD 12142, 3.4 Å resolution)[26] (Table 1). These structures were not present in the AlphaFold2 training set. We tested fitting to densities with or without a 1 Å Gaussian blur applied, and observed comparable convergence to target structures (Supplementary Fig. 1). To emphasize the applicability of our approach to medium-resolution maps, we focused subsequent analyses on fitting to the blurred densities. In each case, an initial known-state model for standard flexible fitting was chosen from an experimental structure determined in a different state than the target density (CLR, PDB ID 7KNT[27]; LAT1, PDB ID 6IRS[28]; ASCT2, PDB ID 6RVX[29]). For CLR, the known-state structure differed from the target-state map in the bending of a single helix; for LAT1, it differed in the arrangement of two neighboring helices; for ASCT2, it exhibited a substantial conformational transition involving most of the transmembrane helices (Fig. 3 - above). Subsequent to density-guided simulations, we used the deposited structure associated with each target state (CLR, PDB ID 6UVA[24]; LAT1, PDB ID 7DSQ[25]; ASCT2, PDB ID 7BCQ[26]) as a reference for fitting quality (Fig. 3 - below). Although such a structure would not be available in an anticipated

application of our approach to a newly reconstructed density, it served as a valuable ground truth for validation of this work; we did not use the RMSD values as a criterion to select the final model from simulations.

For cases where no structure in any alternative state is available, an alternative approach could be to cluster the initial AI-generated ensemble using k-medoids on the basis of pairwise internal distances[30]. In parallel trials of all three of our test cases using this technique (Supplementary Fig. 2), RMSD from the target structure converged to similar values as with our primary approach, indicating this can be a useful alternative in relevant systems. Both k-means and k-medoids clustering appeared robust in enabling selection of starting models for density-guided simulations that converge to low RMSD-to-target: in the cluster producing the final best frame for each test system, the five structures closest to the centroid exhibited a spread in RMSD-to-target of ≤ 0.5 Å (Supplementary Table 1). Both clustering approaches were also effective in seeding distinct conformations. Among the five best-correlated density-guided simulations in each system, the frames with highest compound score sample different parts of structure space: the three different systems exhibit spreads in RMSD-to-target from 1.0 Å to 6.5 Å (Supplementary Table 2). In all cases, when selecting the highest-scoring frame from the simulation with the best mean cross-correlation it produced the lowest RMSD-to-target (or within 0.15 Å from the lowest in one case), indicating that both clustering approaches were effective in distinguishing starting models capable of converging to the target.

On the other hand, AI-based *de-novo* prediction without AlphaFold2 was challenging in all test cases: predictions using the state-of-the-art tool ModelAngelo[8] included only a subset (43−85%) of each protein (Supplementary Table 3), leaving portions of key transitional regions unbuilt (Supplementary Fig. 3). Substantial subsequent user input would be required to complete model building and fitting, in contrast to the final models from our approach, which covered the entire input sequences. Accordingly, we focused the remainder of our study on documenting our approach based on AI-driven generation of a substantial and diverse ensemble, followed by clustering and screening of density-guided simulations to identify representative models capable of fitting to the target state.

**Table 1 | PDB and EMDB IDs for the systems used in the study**

|       | EMDB ID (target state) | PDB ID (target state) | PDB ID (known state) |
|-------|------------------------|------------------------|----------------------|
| CLR   | 20906                  | 6UVA                   | 7KNT                 |
| LAT1  | 30841                  | 7DSQ                   | 6IRS                 |
| ASCT2 | 12142                  | 7BCQ                   | 6RVX                 |

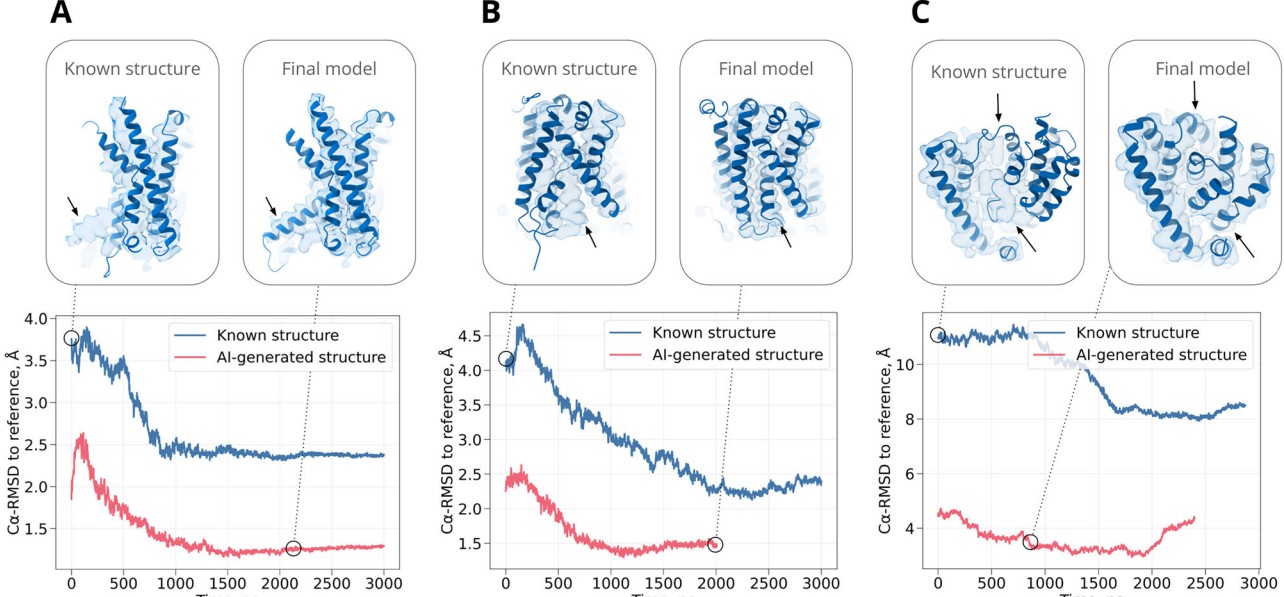

**Fig. 3 | Refinement of an alternative protein state by AI model generation in three test systems. A** CLR, **B** LAT1, **C** ASCT2 refinement results are shown. Plots track Cα RMSD to the target structure during density-guided simulations starting either from the known structure (blue) or the best-fit cluster representative from our generated ensemble (red). Upper insets show the known structure (left) and final model from our approach (right) for each system, aligned to the corresponding density. The arrow highlights the misaligned portion of the protein in the known structure.

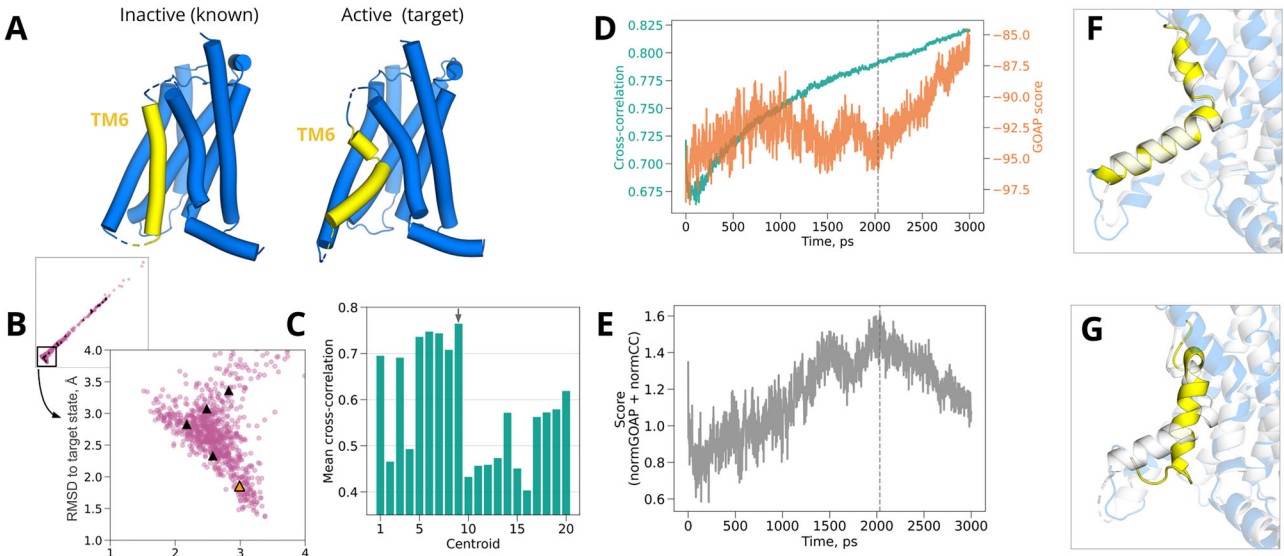

**Fig. 4 | Helix bending in CLR. A** Structures of inactive (left, known state, from PDB ID 7KNT[27]) and active CLR (*right*, target state, from PDB ID 6UVA[27]), with the TM6 helix—where the prominent conformational transition occurs—shown in yellow. **B** Diversity of CLR models from generated ensemble, as indicated by Cα RMSD to target vs. known states. Black triangles highlight the cluster representatives, which were selected as starting points for density guided simulations. The orange triangle marks the representative that achieved the best fit. **C** Mean cross-correlation over density-guided simulations for each cluster representative, with the best-fit representative indicated by a gray arrow. **D** Time-dependent cross-correlations (green) and scaled GOAP scores (orange) during density fitting for the best-fit cluster representative shown in **C**. **E** Time-dependent compound score, combining cross-correlation and GOAP, during density fitting for the best-fit cluster representative. In **D** and **E**, dashed line indicates the frame with the best compound score, selected as the final model. **F** Overlay of the target structure (white) with the model fitted from our ensemble (colored). The bulk of the protein is transparent, with the fitted model in blue; the transitional TM6 region is opaque, with the fitted model in yellow. **G** Overlay of the target structure (white) with the model fitted by standard protocol starting from the known experimental structure (colored). Same coloring scheme is used as in panel *F*.

## Table 2 | Metrics evaluating structure refinement quality

| System name | GOAP score | Cross-correlation | Cα RMSD (region), Å | Cα RMSD, Å | Clashscore | Molprobity score |
|---|---|---|---|---|---|---|
| CLR (generated ensemble approach) | −95.9 | 0.79 | 1.60 | 1.25 | 0.23 | 1.41 |
| CLR (standard approach) | −91.2 | 0.75 | 6.42 | 2.39 | 1.36 | 1.88 |
| LAT1 (generated ensemble approach) | −103.6 | 0.59 | 1.52 | 1.44 | 1.41 | 1.34 |
| LAT1 (standard approach) | −100.2 | 0.52 | 4.08 | 2.33 | 1.98 | 2.01 |
| ASCT2 (generated ensemble approach) | −99.8 | 0.63 | 3.65 | 3.65 | 0.62 | 1.06 |
| ASCT2 (standard approach) | −95.8 | 0.57 | 10.16 | 10.16 | 1.23 | 1.63 |

## Test case 1: helix bending in a 7-TM receptor

We first investigated the characteristic conformational transition of the G-protein coupled calcitonin receptor-like receptor (CLR), a component of both the calcitonin gene-related peptide receptor (CGRPR) and adrenomedullin (AM) receptor, involved in wound healing, vasodilation and other widespread physiological functions[31,32]. A notable conformational transition between functional states of CLR transmembrane domain occurs in the 6th transmembrane helix (TM6, residues 329-354 in the target structure), which is relatively straight in the inactive state without G protein (Fig. 4A—left), but bends upon activation (Fig. 4A—right)[33]. This rearrangement remodels the intracellular binding interface at the N-terminal end of TM6, allowing it to interact with the Gs-protein α subunit. We used an inactive structure of CLR transmembrane domain, extracted from the larger CGRPR (PDB ID 7KNT)[27], as the known state. The cryo-EM density from an active state of CLR, extracted from the larger AM receptor (EMD 20906)[24], served as a target for model refinement.

The bulk of our generated ensemble fell within the cutoff for good quality geometry (Supplementary Fig. 4). Note that deviation from the target state was monitored as an assessment of the ensemble quality, but it is not used for the refinement. After filtering by GOAP score and k-means clustering in MDAnalysis, several (12/20) cluster representative

models produced density-guided simulations with mean cross-correlations over 0.5 (Fig. 4C); a frame within 2.1 ns in the best-fit trajectory had the highest compound score (Fig. 4D, E) and was chosen as a final model.

We then compared our approach to standard flexible fitting starting from the known structure of CLR in an inactivated state. The final model generated in our approach deviated from the activated experimental structure (target state) by only 1.25 Å Cα RMSD (Table 2, Fig. 3A), and included the active-state kink (Fig. 4G), such that even local deviation in TM6 was limited to 1.6 Å (Supplementary Fig. 5). In contrast, the best model from standard density-guided simulations of the inactive structure (known state) deviated 2.39 Å from the target globally, and by 6.42 Å locally, even over 20 independent replicates (Table 2, Fig. 3A, Supplementary Fig. 6). Similarly, another flexible fitting method not based on GROMACS, cryo_fit[34], also did not result in good agreement between the fitted model and the target structure (global RMSD from the target of 3.27 Å after fitting, Supplementary Table 3). Compared to standard known-state fitting, our approach also produced moderately better model quality (−95.9 vs. −91.2 scaled GOAP, 0.23 vs. 1.36 Clash, 1.41 vs. 1.88 MolProbity scores) and fit to the target density (0.79 vs. 0.75 cross-correlation) (Table 2).

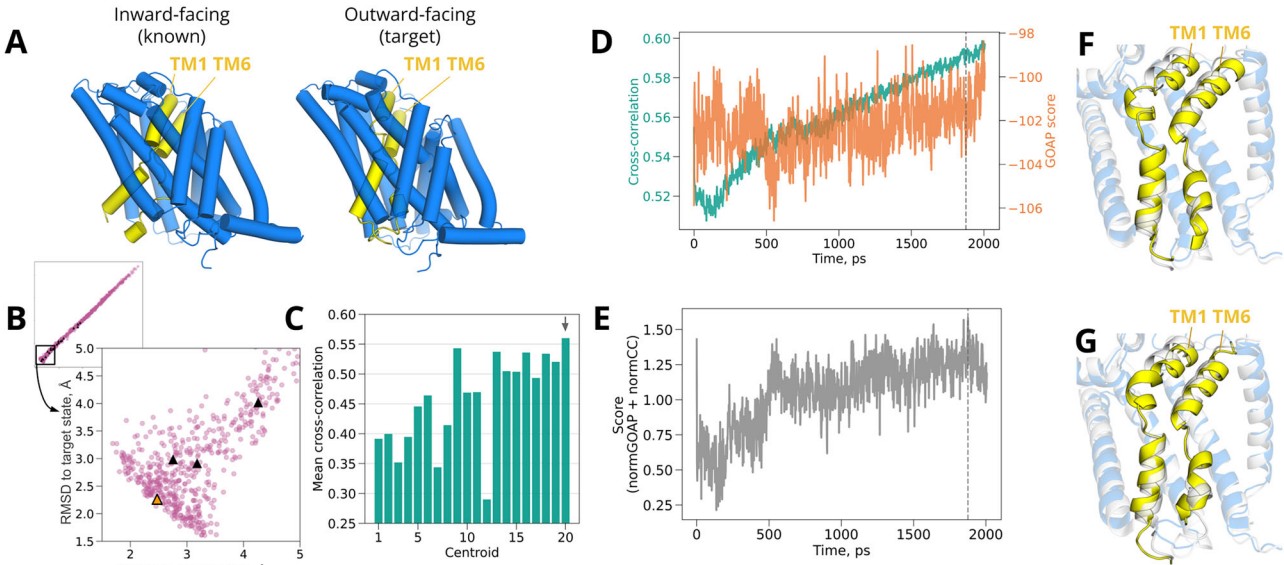

**Fig. 5 | Rearrangement of helices in LAT1. A** Structure of inward-open (left, known state, from PDB ID 6IRS[28]) and inhibited LAT1 (right, target state, from PDB ID 7DSQ[25]), with the TM1 and TM6 helices—where the prominent conformational transition occurs—shown in yellow. **B** Diversity of LAT1 models from generated ensemble, as indicated by Cα RMSD to target vs. known states. Black triangles highlight the cluster representatives, which were selected as starting points for density guided simulations. The orange triangle marks the representative that achieved the best fit. **C** Mean cross-correlation over density-guided simulations for each cluster representative, with the best-fit representative indicated by a gray arrow. **D** Time-dependent cross-correlations (green) and scaled GOAP scores (orange) during density fitting for the best-fit cluster representative shown in C. **E** Time-dependent compound score, combining cross-correlation and GOAP, during density fitting for the best-fit cluster representative. In **D** and **E**, dashed line indicates the frame with the best compound score, selected as the final model. **F** Overlay of the target structure (white) with the model fitted from our generated ensemble approach (colored). **G** Overlay of the target structure (white) with the model fitted by standard density-guided simulations of the known structure (colored). In **F** and **G**, the bulk of the protein is transparent, with the fitted model in blue; the transitional TM1/TM6 region is opaque, with the fitted model in yellow.

Visual inspection of fitted models indicated that the relative success of the generative modeling approach was largely attributable to TM6. The best-fit cluster representative from our generated ensemble, as well as the final model from subsequent density-fitting, included the TM6 kink associated with the active target state (Fig. 4G, Supplementary Fig. 7A, B). Local deviation in TM6 was limited to 1.6 Å relative to the target structure (Supplementary Fig. 5), and key side chains at the intracellular interface were correctly oriented, despite the absence of the Gαs protein binding partner during fitting (Supplementary Fig. 7A, C). Conversely, TM6 remained largely straight after density-guided simulations of the inactive known structure (Fig. 4F), such that local deviation from the target structure in this region was 6.42 Å (Table 2 and Supplementary Fig. 5). Attempted fitting of straight TM6 to the density resulted in partial unwinding of the helix N-terminus, positioning different residues at the G-protein interface than in the target structure (Supplementary Fig. 7B, D). Thus, generated ensemble models offer distinct advantages in the case of a local helix transition between functional states of CLR.

**Test case 2: multiple helix rearrangements in a 12-TM transporter**
To test our approach in context of a larger conformational transition, we next applied it to LAT1, the light-chain component of a heteromeric amino-acid transporter. Together with its heavy-chain binding partner 4F2hc, LAT1 carries hormones and drugs across the blood-brain barrier, and is overexpressed in several cancers[35,36]. A member of solute carrier family 7 (also designated SLC7A5), LAT1 has a 12-TM LeuT fold, and cycles between various inward, occluded, and outward states in a rocking bundle mechanism to transport solutes across the plasma membrane[37]. We used a structure of LAT1 in the inward-open configuration, extracted from the heterodimeric complex (PDB ID 6IRS)[28], as the known state (Table 1). As a target, we used a cryo-EM density (EMD 30841) thought to represent an intermediate between outward-occluded and outward-open states, taken from a complex with the inhibitor 3,5-diiodo-L-tyrosine (diiodo-Tyr)[25]. The principal distinction between the known and target states involved

rearrangement of the kinked helices TM1 and TM6 (residues 47-80 and 240-265 respectively in the target structure) around the diiodo-Tyr binding site (Fig. 5A, Supplementary Fig. 8). The protein density was reported at lower overall resolution than that of CLR (3.4 Å vs. 2.3 Å), offering an additional test case for lower-quality data. We still added 1 Å Gaussian blur to the density.

The results indicated that our LAT1 target density was more challenging than CLR, but could be accurately modeled. Our generated ensemble for LAT1 included more models that exceeded the cutoff for reasonable quality geometry (Supplementary Fig. 4); on the other hand, LAT1 models deviated less than 4 Å (Cα RMSD) from either the known or target (PDB ID 7DSQ)[25] structures, indicating the presence of conformations similar to both states (Fig. 5B). Cluster representatives were successfully fitted using density-guided MD simulations, with 8/20 representatives showing over 0.5 mean cross-correlation (Fig. 5C); a final model was selected within 1.8 ns in the best-fit simulation on the basis of compound score (Fig. 5D, E).

As in the case of CLR, our approach resulted in better fit and geometry metrics than standard density fitting of the known LAT1 structure. The final model based on our generated ensemble was within 1.44 Å Cα RMSD from the inhibited target structure (PDB ID 7DSQ[28]), while the best model based on fitting the inward-open known structure deviated by 2.33 Å for GROMACS density fitting (Table 2, Fig. 3B) or 2.96 Å for cryo_fit tool[34]. Our pipeline model also deviated by only 1.52 Å in the TM1/TM6 region, and was compatible with binding diiodo-Tyr, despite the absence of this inhibitor model and its density during fitting (Table 2, Supplementary Fig. 8). Conversely, the known-state fitted model deviated over 4.08 Å (Table 2) in the TM1/TM6 region, and oriented a Tyr residue to clash with the anticipated inhibitor pose. Compared to standard known-state fitting, our generative ensemble setup produced modestly better model quality (−103.6 vs. −100.2 scaled GOAP, 1.41 vs. 1.98 Clash, 1.34 vs. 2.01 MolProbity scores) and fit to the target density (0.59 vs. 0.52 cross-correlation). Thus, our approach facilitated modeling in the context of a multi-helix conformational transition, including a ligand binding site in a functional intermediate.

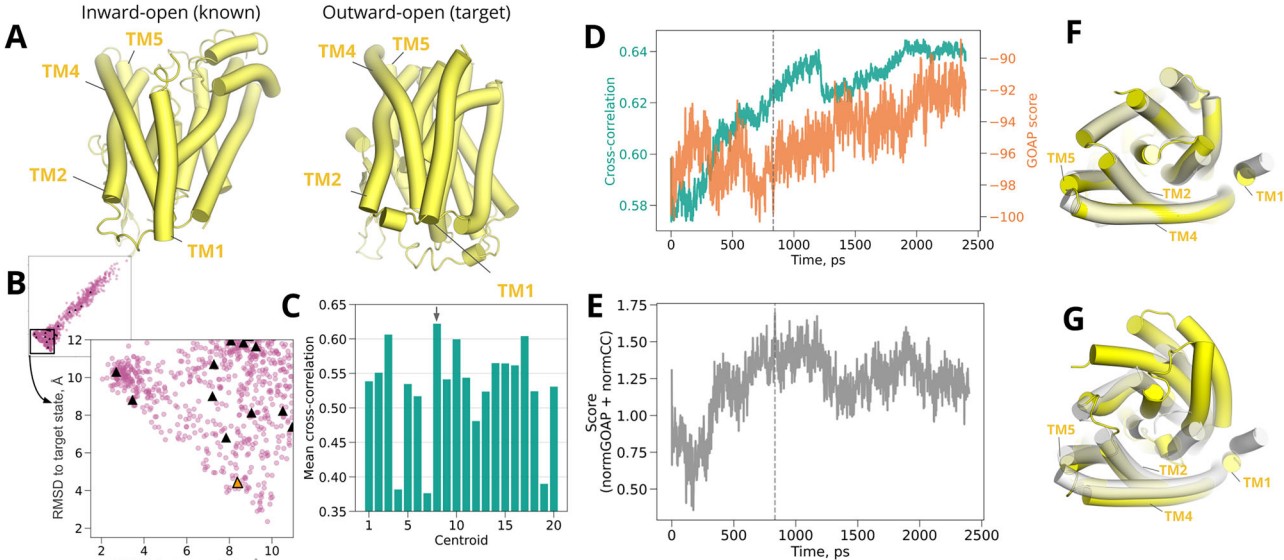

**Fig. 6 | Domain rearrangement in ASCT2. A** Structures of inward-open (left, known state, from PDB ID 6RVX[29] and outward-open ASCT2 (*right*, target state, from PDB ID 7BCQ[26]). **B** Diversity of ASCT2 models from generated ensemble, as indicated by Cα RMSD to known vs. target states. Black triangles highlight the cluster representatives, which were selected as starting points for density guided simulations. The orange triangle marks the representative that achieved the best fit. **C** Mean cross-correlation over density-guided simulations for each cluster representative, with the best-fit representative indicated by a gray arrow. **D** Time-dependent cross-correlations (green) and scaled GOAP scores (orange) during

density fitting for the best-fit cluster representative shown in C. **E** Time-dependent compound score, combining cross-correlation and GOAP, during density fitting for the best-fit cluster representative. In **D** and **E**, dashed line indicates the frame with the best compound score, selected as the final model. **F** Overlay of the target structure (white) with the model fitted from our generated ensemble approach (yellow). **G** Overlay of the target structure (white) with the model fitted by standard density-guided simulations of the known structure (yellow). In panels *F* and *G*, helices contributing to the scaffold domain are labeled; remaining helices constitute the transport domain.

## Test case 3: domain rearrangement in an 8-TM transporter

To test the applicability of our approach to a more distributed transition between known and target states, we investigated ASCT2, a homotrimeric member of solute carrier family 1 (SLC1A5) with an 8-TM GltPh fold[38]. This transporter is widely expressed in human tissues, and upregulated in numerous cancers, making it an important target for drug development as well as biophysical characterization[39]. ASCT2 is structurally and mechanistically distinct from LAT1, consisting of a scaffold domain comprising helices TM1-2 and TM4-5, and a transport domain comprising TM3 and TM6-8 along with two interhelical hairpins[40]. Transport occurs by a one-gate elevator mechanism, in which the transport domain moves between inward and outward states relative to the scaffold[41]. We used a structure of an ASCT2 monomer in an inward-open configuration, extracted from the trimeric complex (PDB ID 6RVX)[29], as the known state (Table 1). As a target, we used a cryo-EM density (EMD 12142) determined to similar resolution as LAT1 (3.4 Å) in an outward-open configuration, taken from the homotrimeric complex in the presence of the inhibitor 4-(4-phenyl-phenyl)carbonyloxypyrrolidine-2-carboxylic acid (Lc-BPE)[26]. Conformational cycling in this system involved evident rearrangements among all helices and hairpins; accordingly, we considered the entire monomer as a transition region (Fig. 6A).

Despite its relatively distributed transition between known and target states, our approach was similarly successful in modeling ASCT2 as previous test cases. Our generated ensemble included an intermediate fraction with low quality geometry (Supplementary Fig. 4); however, models within 2 Å Cα RMSD of the known structure deviated at least 10 Å from the target (PDB ID 7BCQ[26]) and vice versa, consistent with substantial rearrangements observed between the two states (Fig. 6B). Most (16/20) cluster representatives could be fit with mean cross-correlation to the target map over 0.5 (Fig. 6C), with the final model selected within 0.8 ns in the best-fit simulation (Fig. 6D, E).

Fitting of ASCT2 was substantially more accurate using our ensemble approach than starting only from the known structure (Fig. 3C). The final model based on our generated ensemble was within 3.65 Å Cα RMSD from

the outward-open target structure, while the best model based on fitting the inward-open known structure deviated by 10.16 Å for GROMACS density fitting or 11.73 Å for cryo_fit tool (Table 2 and Supplementary Table 3). The overall RMSD value converged to similar values with or without blurring the original cryo-EM maps during our simulation in the ensemble approach Supplementary Fig. 1. When aligned on the scaffold domain (TM1-2, TM4-5), all transmembrane helices in our final model were largely superimposable with the target structure (Fig. 6F); for the fitted known state, rearrangements were evident in helices comprising the transport domain (Fig. 6G). As in previous cases, our approach also produced modestly better model quality (−99.8 vs. −95.8 scaled GOAP, 0.62 vs. 1.23 Clash, 1.06 vs. 1.63 MolProbity scores) and fit to the target density (0.63 vs. 0.57 cross-correlation). Thus, generated ensemble model fitting appeared particularly useful in the context of a conformational transition across multiple transmembrane helices.

Interestingly, the initial ensemble for ASCT2 included a model with a modestly better RMSD relative to the target structure (2.35 Å) than the final result from our *k*-means-based pipeline (3.65 Å) (Fig. 6), or from our alternative k-medoids clustering (3.46 Å, Supplementary Fig. 2). As shown in previous sections, this issue did not appear with CLR or LAT1 where the best member of the starting ensemble included a model with 1.37 Å and 1.44 Å RMSD, respectively. After running our pipeline, our final models deviated equally or less from our *k*-means-based pipeline (1.25 Å and 1.44 Å RMSD, respectively), and from our alternative k-medoids clustering (1.46 Å and 1.64 Å, respectively) (Figs. 4 and 5 and Supplementary Fig. 2). Although in the case of ASCT2 our clustering method appeared to exclude a particularly optimal model, this hit could not have been identified given our presumed advance knowledge of only the target density, not the target structure. Alternatively, we attempted to select models approximating the target structure directly from the filtered ensemble, prior to clustering or simulations, on the basis of cross-correlation to the target density; however, the best-correlated models for ASCT2 were visibly different, and exhibited relatively high RMSD (9.57-11.39 Å), with respect to the target structure (Supplementary Fig. 9). Thus, knowledge of the target density was

insufficient to identify plausible models prior to clustering and simulations, supporting the overall utility of our approach.

Motivated by challenging characteristics particularly of the ASCT2 test case, we also evaluated alternative approaches to selecting a final model. Although GOAP is an established metric for model quality particularly for simulations in GROMACS, MolProbity is a common alternative in structural biology[42]. Monitoring MolProbity vs. GOAP scores during density-guided simulations resulted in final models with similar RMSDs relative to the target structure (3.39 Å vs. 3.65 Å respectively), indicating either metric could be effectively applied. Perhaps more critically, our primary approach relied on a compound score based on cross-correlation and GOAP scores normalized within the best-correlated density-guided simulation; an alternative strategy would be to normalize each value across all simulations of a given system. For CLR and LAT1, such global normalization selected precisely the same frame as our primary approach (Supplementary Fig. 10). For ASCT2, global normalization selected a model from a different trajectory than our primary approach, with a modestly better GOAP score but worse cross-correlation. Notably, RMSD with respect to the target structure was 8.26 Å for the final model using global normalization, vs. 3.65 Å when normalizing within each trajectory, indicating that our approach yields an equivalent or better model in all our test cases.

## Discussion

In this study, we propose and implement an approach that leverages AI generation of protein models along with density-guided MD simulations for atomistic modeling into cryo-EM maps. Automated structure determination can in some cases be achieved by rigid or flexible fitting of known structures[9–14]. However, each of our test systems illustrates a challenging case where standard flexible fitting in GROMACS or alternative cryo_fit implementation based on a known structure fails to accurately refine a complex target in a substantially different state. Activation by G-protein binding to CLR involves bending of a key transmembrane helix; inhibition by diiodo-Tyr locks LAT1 in an intermediate, with two transmembrane helices rearranged relative to the inward-open state; and conformational cycling between inward-open and outward-open states involves distributed remodeling across ASCT2, particularly in the transport relative to scaffold domains. The success of our approach in these systems indicates its applicability to multiple dynamic proteins, including physiologically critical membrane transporters and receptors. It is likely transferrable to a variety of proteins that undergo conformational transitions, though current limitations in AlphaFold2 restrict it use in more complex biological assemblies, for example involving nucleic acids; fortunately, AlphaFold3[43] has shown potential for modeling such systems.

In our test cases, standard density-guided MD simulations based on a single known structure produce models that deviate from the target structure by 2.33-10.16 Å, even when replicated up to 20 times for a single system. In contrast, flexible fitting using a clustered subset of full-length AI-generated models as simulations starting points recapitulated target structures within 1.25-3.6 Å. Python scripts are available for several key steps in our approach, which could likely be further automated in future. Manual tools such as Phenix or Coot might further improve model quality, though we deliberately avoided such additional steps in order to optimize our approach for cases of limited structural biology or system-specific expertise. Notably, this approach removes the need to modify initial models for density fitting, for example by building heavy atoms or residues unresolved in a previous experimental structure, since AlphaFold2 predicts positions for all heavy input atoms.

Although this approach may prove less successful for systems with limited structural data, all our target systems were excluded from the AlphaFold2 training set, and in fact have been used in previous work to validate the applicability of this tool to new systems[19]. In cases for which restricted homology or intrinsic disorder prevents approximation of the target conformation[44], multi-step approaches using progressive resolution[45], gradual parameter changes[12] or enhanced sampling[10,11] may prove valuable. Similarly, we adopted GOAP score as an assessment of geometric quality

based on previous work[22] and straightforward implementation in GROMACS density-guided simulations. However, GOAP scoring is based on a reference set of > 1000 divergent structures resolved to < 2 Å[21], and may be accordingly biased towards characteristics of soluble proteins determined by X-ray crystallography; in some cases, alternative metrics such as MolProbity score[42] may be preferred.

Parameters for ensemble generation were selected based on previous work[46], while parameters such as GOAP-score cutoff and number of clusters were pre-determined on the basis of accessible resources. Alternative parameter values might enhance performance particularly in other systems, but were kept consistent in this work in order to assess applicability to our parallel test cases. As illustrated by the wide range of structures generated in all test cases (Supplementary Fig. 11), a critical challenge in our approach was to select a subset of starting models for density-guided simulations that would retain sufficient diversity to identify even a poorly sampled state, yet be tractable given finite time, compute and user resources. Although in one case (ASCT2) the initial ensemble included a model with a modestly better target-structure RMSD than the final result from our pipeline (Fig. 6 and Supplementary Fig. 2), this hit could not have been identified in the anticipated experimental case where the target structure is unknown. Increasing the number of initial models or clusters could improve the likelihood of capturing states closer to the target, but at a steep computational cost. We suggest that future work should focus on enhancing AI sampling strategies to better explore biologically relevant conformational spaces, and improving clustering methods that balance diversity with proximity to functional states. Within the scope of presently accessible tools, we expect our work to represent a particularly relevant approach in cases of limited structural-biology or system-specific expertise; for comparison, the target structure of CLR was determined using Rosetta and MDFF plus several rounds of manual fitting in Coot and real-space refinement using Phenix[24].

Our approach further leverages stochastic subsampling of MSA depth to generate more diverse initial models, allowing us to sample conformations suitable for automated fitting to the target state. Some de-novo methods of cryo-EM structure determination, such as those implemented by DeepMainMast[47] and similar tools[48–58], also incorporate prediction tools like AlphaFold or RoseTTAFold to generate protein or fragment models. However, even with AI-generated models, predicting divergent or rare protein conformations may be challenging[59]. A distinctive feature of the approach in this work is the clustering of a substantial and diverse ensemble of full-length models into an accessible subset of broadly distributed models. Thus, our approach might be particularly applicable to proteins with multiple metastable states or otherwise flexible features. Whereas MSA subsampling proved effective in our test cases, other recent approaches to generating diverse structural ensembles in AlphaFold2[60,61] or AlphaFold3[43] may offer further advantages. In particular, sequence clustering or alignment alterations may improve sampling or allow tuning to specific targets or transitions. Although the alphafold2_conformations package used here does not support oligomeric proteins[19], tools such as AlphaFold2-multimer may enable the application to larger, more complex systems[62]. Alternative approaches to conformational sampling include extensive MD simulations, but may be substantially more resource intensive than the generative modeling approach taken here.

Integrating flexible fitting with ensemble generation offers further potential advantages[63,64]. Whereas classical de-novo methods[5–7] work reliably with high-resolution cryo-EM data, they may be less applicable to medium- and low-resolution maps. In such cases, density-guided MD simulations may be particularly valuable in integrating physics-based force fields with cryo-EM data. In fact, higher resolution densities are often blurred for the purpose of flexible fitting, reducing barriers to translocate model regions from one density area to another[10]. Target densities tested in this work were resolved to 2.3-3.4 Å overall, and were subjected to an additional 1 Å Gaussian blur prior to fitting. This blurring did not affect the final model quality; simulations using either original or blurred maps converged to similar RMSD values relative to the target structures

(Supplementary Fig. 1), consistent with indications that modest differences in map resolution have limited impact in context of a well matched initial model. This observation also supports the applicability of our approach even when only moderate resolution maps are available.

Furthermore, reducing our ensemble to cluster representatives allowed us to run only 20 density-guided simulations for each system, while representing the diversity of 1250 models. Even where the chosen model may not have represented the best model, application of the MD force field during flexible fitting evidently enabled us to optimize geometry alongside map correlation. We were able to generate and fit models for each system with relatively high accuracy despite the absence of secondary structure restraints or membrane mimetics, further streamlining the protocol. Manual verification of the resulting models might be appropriate prior to final deposition. Although our test systems were extracted from larger complexes, flexible fitting in the absence of such partners nonetheless recapitulated key interfaces with accessory G proteins (for CLR) or inhibitors (for LAT1). Thus, our approach offers a straightforward integration of the strengths of both model generation and flexible fitting, applicable even to membrane-protein systems representing a range of data quality and binding partners.

## Methods

### Test systems

Three different systems, each including cryo-EM structures in at least two different conformations, were selected as known and target states for pipeline validation: inactive and active conformations of CLR transmembrane domain[24,27], inward-open and inhibited conformations of LAT1[25,28], and inward-open and outward-open conformations of ASCT2[26,29]. All target structures and densities were released after the training dataset for AlphaFold v.2.0.1. To assess the applicability of *de-novo* model generation in these systems, we also attempted to predict structures based on their target densities in ModelAngelo[8]. We used the default ModelAngelo build command for the case where both the cryo-EM map and the protein sequence were available, following the instructions provided in the ModelAngelo GitHub repository (https://github.com/3dem/model-angelo).

### Ensemble generation

AlphaFold2 models were generated with the alphafold_conformations package[19]. Before ensemble generation, we assessed per-residue prediction quality for each protein sequence using the AlphaFold Protein Structure Database[65]. Terminal residues with low-confidence predictions (pLDDT scores below 70) were trimmed from the sequence to improve further modeling reliability and exclude regions unlikely to correspond to well defined densities. MSAs were obtained from the MMseqs2 database[66]. The MSA depth was subsampled according to previously proposed methods[19], using a single recycle and no energy minimization. As optimized in previous work[46], we used MSA depths in the range of 5-30 to generate 50 models at each depth, resulting in 1250 initial models for each system. AlphaFold2 inference was executed on an NVIDIA DGX A100 GPUs, requiring ~40 hours per test case.

### Model clustering and rigid alignment

After model generation, we filtered out models that were completely or partially misfolded on the basis of GOAP score[21] scaled with respect to protein sequence length. GOAP source code was downloaded from https://sites.gatech.edu/cssb/goap/. We discarded models with scaled scores above −100, a round threshold value selected arbitrarily, but which proved to filter models with the worst geometry in all three test cases (Supplementary Fig. 4). The remaining models were aligned to the structure assigned as the already-known state using PyMOL[67], and clustered using the k-means method on the basis of Cα distances in the MDAnalysis Python package[20]. Evaluation of 20 cluster representatives for each system was determined in advance to be accessible given the available compute resources; this number evidently enabled sufficient structural diversity to improve fitting in all our test cases, within practical limits of reproducibility. Rigid alignment of cluster centroids to the cryo-EM density assigned as the target state was

performed using the rigidbodyfit 1.2.1 package available at https://gitlab.com/cblau/rigidbodyfit.

To evaluate the extensibility of our approach to systems where no structure is known for any state, we also executed a parallel analysis of each AI-generated ensemble without initial alignment to any known state, directly computing a distance matrix representing RMSD in pairwise internal distances between all Cα atoms. We then used this matrix as input to the k-medoids clustering algorithm[30], which can be applied to arbitrary distance matrices, potentially more suitable for clustering on structural dissimilarity. This approach addresses potential limitations of our original approach using k-means clustering on aligned Cartesian coordinates, which could be overly biased by the known structure or otherwise fail to meaningfully capture structural similarity. In all three of our test cases (Supplementary Fig. 2), RMSD from the target structure converged to similar values with or without known-state alignment, indicating that either approach can be successfully applied.

Clustering ran on a local cluster using one x86 CPU node within minutes. Python scripts for model clustering and rigid alignment are available on Zenodo (https://doi.org/10.5281/zenodo.14749349).

### Density-guided simulations

Initial preparation of the target density was performed in UCSF Chimera[68]. Preparation included segmentation of the original density map and manual picking of the segments corresponding to the protein of interest. After merging selected regions, to explore the applicability of our approach to medium-resolution maps, we added a 1 Å Gaussian blur to each target density; however, we noted that RMSDs converged to similar values with or without blurring (Supplementary Fig. 1). Missing loops in the known experimental states of CLR and LAT1 were built using MODELLER[69] in UCSF ChimeraX[70].

System preparation, equilibration and density-guided simulations were performed in GROMACS 2023.1[71]. Each simulation ran on a local cluster using two x86 CPU cores and one NVIDIA GeForce RTX 2080 GPU. For each test case, we ran 20 systems in parallel with a total wall-clock time of 10-20 hours. TIP3 water, the neutralising amount of NaCl, and the CHARMM36 force field[72] were used for all simulations. It has been shown that embedding in lipids or detergents does not substantially improve density-guided simulations of a membrane protein[22], so for simplicity they were not included. Bonded and short-range nonbonded interactions were calculated every 2 fs, and periodic boundary conditions were employed in all three dimensions. The particle mesh Ewald (PME) method[73] was used to calculate long-range electrostatic interactions. A force-based smoothing function was employed for pairwise nonbonded interactions at 1 nm with a cutoff of 1.2 nm. The temperature was maintained at 300K with the velocity-rescaling thermostat[74]. The pressure was kept at 1 bar with the c-rescale barostat[75]. Forces from density-guided simulations were applied every $N = 2$ steps. A scaling force factor of $10^3$ kJ/mol was combined with adaptive force scaling in GROMACS, as the latter ensures a steady and automated increase in similarity between simulated and reference densities; simulations terminate automatically when overall forces on the system are too large to be compatible with the integration time step. All simulation setup parameters and preparation scripts have been made available on Zenodo (https://doi.org/10.5281/zenodo.14749349). Density fitting tutorial in GROMACS is available at https://tutorials.gromacs.org/density-fit-simulation.html.

To assess the performance of alternative flexible fitting methods on our known experimental states, we used cryo_fit[34], an automated flexible fitting tool integrated into the Phenix software suite[76]. All default options were used, except the "Multiply EM weight by this number" field, which was set to 100 to ensure that the fitting force was sufficiently strong to balance the MD force field and stereochemistry preserving constraints.

### Analysis

For each target, we identified the simulation with the highest mean cross-correlation to the target density, then selected the frame with the highest compound score as a final model. Standard cross-correlation calculations

were performed using a modified GROMACS version, available at https://gitlab.com/gromacs/gromacs/-/tree/ml_fscavg_main. The compound score was defined as a combination of the GOAP score and the cross-correlation, enabling optimization of protein geometry as well as map fit. First, both the cross-correlation and the GOAP score absolute value were subjected to min-max normalization over the interval [0, 1]. The final compound score was then calculated as the sum of the normalized GOAP score and the normalized cross-correlation. Phenix MolProbity clash scores and total scores were calculated for additional model quality assessment[42]. RMSD was calculated in PyMOL[67]. Python 3.9 scripts were utilised to combine pipeline steps, conduct analyses, and generate figures. Deviation in regions of local conformational change was calculated for residues 329-354 for CLR, and for residues 47-80 and 240-265 for LAT1; for ASCT2, the entire polypeptide was consider the region of change.

## Data availability

All data needed to evaluate the conclusions are present in the paper or the Supplementary Information. The generated models with ChimeraX sessions, density-guided simulation input files and trajectories can be found on Zenodo (https://doi.org/10.5281/zenodo.14749349). A tutorial for the density-guided simulations performed in GROMACS is available at https://tutorials.gromacs.org/density-fit-simulation.html.

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

## Acknowledgements

Computational resources for simulations were provided by the National Academic Infrastructure for Supercomputing in Sweden (NAISS 2024/3-49). AlphaFold2 runs were performed using the computing facilities of the Berzelius through NAISS (grant no. Berzelius-2023-244). Protein emoji used in the pipeline scheme is taken from https://github.com/whitead/protein-emoji, CC-BY 4.0. This work was partially funded through a Marie

Sklodowska-Curie Postdoctoral Fellowship 101107036 to NH and grants from the Swedish Research Council (VR; 2019-02433, 2021-05806), the Knut and Alice Wallenberg foundation (KAW; 2023.0254) and the BioExcel-3 Centre-of-Excellence (EuroHPC Joint Undertaking; 101093290) to E.L.

## Author contributions

Conceptualization: N.H., T.S., R.J.H., E.L. Data curation: T.S., N.H. Formal analysis: T.S. Funding acquisition: N.H., E.L. Investigation: T.S. Methodology: T.S., N.H. Project administration: N.H., R.J.H., E.L. Resources: N.H., E.L. Software: T.S., N.H. Supervision: N.H., R.J.H., E.L. Validation: T.S. Visualization: T.S. Writing—original draft: T.S. Writing—review and editing: T.S., N.H., R.J.H., E.L.

## Funding

## Competing interests

The authors declare no competing interests.

## Additional information

**Peer review information** : *Communications Chemistry* thanks the anonymous reviewers for their contribution to the peer review of this work. Peer review reports are available.

