## [Transparent Peer Review file · Communications Chemistry]

Modeling cryo-EM structures in alternative states with AlphaFold2-based models and density-guided simulations

Corresponding Author: Dr Nandan Haloi

Version 0:

Reviewer comments:

Reviewer #1

(Remarks to the Author)

The paper by Tatiana Shugaeva et al. suggests an automated pipeline to obtain atomic protein coordinates for a given cryo-EM density map: Candidate models are generated with AlphaFold2 using subsampling of multiple sequence alignments and subsequently filtered via the GOAP score. Next, k-means clustering is used to sample 20 representative structures, which are then steered onto the target density with density-guided MD simulations. A final model is selected from these simulations based on cross-correlation and GOAP score.

The suggested method seems very useful for proteins with multiple conformational states and is applicable to density maps with low to medium resolution. Moreover, the authors illustrate a clear improvement over a more standard approach (reference 14), where a single known atomic structure is steered onto the target density (alternative conformation). The proposed method is able to accurately model challenging local regions undergoing conformational changes. Overall, the idea of the method and the individual steps are clearly described and well structured. Additionally, the figures are well-designed (in particular Fig 3, 4, 5, 6, S3, and S4) and convey the most relevant information.

However, the following issues, listed in order of priority, must be resolved before publication:

1. k-means clustering is one of the most central points of the proposed method. However, it has multiple issues:

1.1 The candidate structures are initially aligned to the known structure. What if no known structure exists? This currently restricts the applicability of the method.

1.2 MDAnalysis is used for k-means clustering of the candidate structures. Thus, the cartesian coordinates are flattened into a single feature vector, and subsequently, the KMeans class from sklearn with the Euclidean distance is used for the actual clustering. The large number of features (number of C α * 3) may already be a problem. Moreover, the distances between candidates are not meaningful as they are based on the alignment to the known structure. I suggest using RMSD (or IDDT, GDT, QCS, ...) as a distance measure between candidates, then calculate the distance matrix, and directly use it for the KMedoids class (not KMeans as it does not allow for a distance matrix and relies on the Euclidean distance) of sklearn. This would also solve the problem of 1.1 as no alignment to the known structure would be necessary.

1.3 Candidates closest to the cluster centroids are selected. This approach is valid. However, Fig 4B, 5B, and 6B indicate that AlphaFold2 generated candidates with an RMSD less than 0.5, 1.3, and 2.0 angstroms to the target structures CLR, LAT1, and ASCT2, respectively. This is a lower RMSD than the best k-means candidates even after the density-guided MDs (which according to Fig 3 lead to further improvements over the AI-generated structures). From this, I conclude that k-means is not selecting optimal candidates (maybe due to points 1.1 and 1.2, and because k-means will not select the most diverse structures, but averaged structures for each cluster). Solving the max-min diversity problem might be an approach to selecting better candidates. Additionally, marking the selected 20 candidate structures in Fig 4B, 5B, and 6B would be helpful to evaluate the sampling method.

1.4 The authors correctly write 'and used the model closest to the cluster centroid for fitting' (not the centroid itself), but later in the article, the authors refer to the cluster representatives as the cluster centroids, which is not correct. This is also the case for Fig 2.

2. The density-guided simulation with the highest mean cross-correlation is selected and the frame with the highest compound score is chosen as the final model. Why is a selection of the best simulation necessary? Why not directly determine the frame with the highest compound score across all density-guided simulations? If the goal is to obtain a structure that optimally fits the cryo-EM density while maintaining high geometric and chemical quality, I believe this approach is more direct. The proposed method potentially excludes simulations that start with a low compound score, but progress to very high scores over the course of the simulation. Perhaps the authors are concerned about selecting unstable models. To address this, the first 500 or 1000 ps of the density-guided simulations could be excluded, or a more sophisticated analysis could be implemented.

3. The results suggest a clear improvement (RMSD) and slight improvements in model quality compared to a more standard approach (reference 14). However, it also requires significantly more runtime. Generating 1250 models via AlphaFold2, filtering based on the GOAP score, clustering, and 20 density-guided MD simulations compared to a single MD simulation with an equal amount of steps. I did not find this comparison in the article. While Fig 3 suggests that the standard method will not benefit from longer simulation times, maybe 20 simulations from the same starting structure with different starting velocities would provide better results? A discussion on this topic would be helpful.

4. According to Fig 3 the density-guided simulations were performed with different number of steps for the three test systems, namely 3000, 2000, and approximately 2400 for CLR, LAT1, and ASCT2, respectively. Were the 20 simulations within one test case performed with the same number of steps? What was used as a stopping criterion? A clearer description is necessary here.

5. How were the parameters of the suggested **automatized** method selected? For instance, the number of initially generated models by AlphaFold2, the threshold for the GOAP score in filtering, or the number of clusters for k-means. If they were selected based on all three test cases, they no longer serve as true test cases since they were used for parameter optimization. Additional unseen test cases would then be necessary.

6. Minor issues and suggestions:

- Filtering with the GOAP scores is not shown in Fig 2.

- Fig 1 lacks a precise description: 1. Differentiation between top and bottom is missing. 2. Descriptions of the arrows pointing to the yellow regions are missing. 3. Bottom: The density is sometimes smaller than the atomic protein. 4. Bottom: Labelling of the arrows - for example with 'Steered MD' - is missing. 5. The authors write 'based on a previously known structure (ribbons) may be challenging for a given protein (blue)'. Ribbons and blue both refer to the same protein? Also, everything (proteins + densities) is blue in the figure.

- Fig 2 has rectangles around the protein emojis, which can be easily mistaken for the density. Thus, I suggest to remove all rectangles from the figure.

- Fig 3 depicts two structures for A, B, and C. Without reading the text, it is unclear what the structures represent. Adding text like 'known structure' and 'final model' to the figure would be helpful.

- Fig 4, 5, and 6 are great. They visualize the key results exceptionally well!

- Table 2 provides an excellent overview of the evaluation criteria and results. However, the corresponding description in the main text is somewhat imprecise. For example, the authors write 'our generated ensemble was within 3.2 angstroms C α RMSD from the outward-open target structure, while the best model based on fitting the inward-open known structure deviated more than 10.1 angstroms' to describe the deviations of 3.14 angstroms and 10.16 angstroms. I suggest using the exact values to match Table 2 and avoid unnecessary comparisons with it.

- It is initially unclear which clustering algorithm was used. This should be mentioned directly in the abstract and at the beginning of the results section under 'We then clustered these to identify...'.

Reviewer #2

(Remarks to the Author)

Shugaeva et al. present a refinement approach for modeling atomic coordinates into cryo-EM maps, combining generative AI models with density-guided molecular dynamics simulations. While the study demonstrates improved fitting accuracy for three membrane proteins undergoing conformational transitions, several aspects warrant critical consideration regarding novelty, methodology, and scope.

The authors state in the abstract that their method combines "generative AI" with simulation-based refinement to address challenges in building alternative states of membrane proteins. However, the generative component relies entirely on AlphaFold2, a pre-existing software not developed by the authors. As such, referring to it as "Modeling cryo-EM structures in alternative states with AlphaFold2 based models and density-guided simulations" would be more accurate. Although, the stochastic subsampling of MSA depth within AlphaFold2, as described in the Results section ("we first used stochastic subsampling of input MSA depth in AlphaFold2...") is admittedly a novel approach. Similar services, such as DeepMainMast (Terashi et al., *Nature Methods* 2024), already offer cryo-EM model refinement using AlphaFold2 structures and density information. It should be more explicitly clarified in the manuscript what distinguishes the authors' implementation from these existing services.

The rationale for structure-based clustering after generating AlphaFold2 models is not fully clear to me. If the goal is to identify structures closer to the target map, it seems logical to directly initiate MD simulations from the AlphaFold2 structure exhibiting the highest initial correlation with the target density. The added complexity of clustering and selecting centroid structures requires some further explanation.

I also believe the choice of the Generalized Orientation-Dependent All-Atom Potential (GOAP) score as a structure geometry metric requires further explanation. The authors should provide a comparative analysis of GOAP against other established metrics like MolProbity, Q-scores, Ramachandran outliers, or clash scores. A discussion of GOAP's strengths and weaknesses, as well as its sensitivity to specific structural features, would enhance the validity of the results.

In the discussion, the authors mention their method's applicability "among other things to multi-state proteins" (page 13).

This statement is somewhat vague and should be clarified with specific examples. Additionally, the limitations of the method, ex., concerning nucleic acids, nucleic acid-protein complexes, or systems with limited structural data, should be more explicitly stated.

The authors assert that "flexible fitting may fail to accurately refine a complex target for which the initial model substantially differs" (page 13). However, they do not provide convincing evidence that their method outperforms established flexible fitting methods, in particular those specifically designed for such scenarios. A comparison against benchmarked methods like the one described in reference 12 of the manuscript (Igaev et al., *eLife* 2019), possibly using the same benchmarks found in that reference, would greatly help to clarify the issue.

The validation strategy relies on comparing the generated models to experimentally determined structures of the target state. However, I see it, the method essentially uses AlphaFold2 to identify a state closer to the target, followed by standard density-guided MD simulations. Any method capable of generating a similar starting point would likely achieve comparable results. The manuscript should address this by demonstrating that the method outperforms alternative approaches for generating suitable starting models, such as automated fitting procedures in software packages like Phenix, Isolde, or Rosetta.

Overall, I feel the manuscript lacks a clear focus regarding its intended application. Is the method primarily aimed at flexible fitting, geometry optimization, modeling distant states, or ensemble refinement? The data presented and the validation strategy are insufficient to support its effectiveness across all these applications. To position the work more clearly, the authors should clarify in the introduction which specific problems they are trying to address and then provide corresponding benchmarks against state-of-the-art methods for those problems throughout the manuscript.

While the presented approach shows promise for modeling cryo-EM structures in alternative states, the manuscript requires further work to address concerns regarding novelty, methodology, validation, and scope. A more rigorous comparison to existing methods, a clearer articulation of the method's strengths and limitations, and a more focused validation strategy are essential to establish the true value of this work.

Reviewer #3

(Remarks to the Author)

Shugaeva et al. present a novel approach for fitting atomic models to cryo-EM maps based on a recently established approach for generating models in an ensemble of multiple conformations with AlphaFold2 combined with density-guided simulations. The paper is well-written and shows that this approach seems very promising in cases where atomic modelling is challenging as demonstrated for 3 case studies from the EMDB.

However, the approach seems to be rather manual at this point (with several independent steps including a manual analysis one for selecting fitting trajectories and best frames) and would greatly benefit from automation to make it reliably usable. There is some mention right at the end of the methods that "Python 3.9 scripts were utilised to combine pipeline steps, conduct analyses and generate figures" but it would be great if these were described throughout and made available with instructions. Some of these seem to be available in the Zenodo record but this is not completely clear. Providing a complete user-friendly implementation would make this study much more valuable to the general structural biology community.

I also have several more specific comments.

Major

- The current manuscript does not discuss the difficulty of using this approach for non-expert users, which seems to be considerable. There is a mention in the discussion that the authors implement an approach but otherwise we only see brief descriptions of the steps outside of the methods section, which seems rather involved with several packages that need to be installed and used.
- The authors mention in the end of the first section of the results that they applied a Gaussian blur to the target densities to mimic lower resolution situations. However, there is no mention of the impact of this blurring in the case study results. Please include an additional supplementary figure for each case study comparing the results with and without the blur. It can often be the case that density-guided simulations struggle in higher resolution cases due to the more rugged optimization landscape. It may well be that this is not the case here as the best models from the ensemble are already very close to the target and this would be good to show. On the other hand, if the authors believe that this Gaussian blur is useful for better fitting then this should be included as part of the approach. There is an indication of this in the discussion.
- While this approach outperforms the standard density-guided simulation approach with a single model that is far from the target density when using their all-atom density-guided simulation implementation in Gromacs, no comparison is made to any other method, such as MDFF, normal mode-based flexible fitting, Rosetta or traditional real space refinement with Phenix.

- The difficulty of obtaining the target models in the PDB that were used for comparison and the need for the approach should also be discussed in comparison to the proposed approach. In particular, the CLR system required Rosetta and MDFF as well as several rounds of manual fitting with Coot and real space refinement with Phenix.
- The 2nd and 3rd case studies still show fairly low cross-correlations, and the 3rd case still has an RMSD worse than 3 Å, even with the proposed approach. Have the authors tried refining the output models further with standard approaches and is this successful in improving them? This would be important to confirm that the solutions are not stuck in local minima.
- The authors always pick the simulation with the highest mean cross-correlation but in many cases several simulations have similar mean cross-correlation. Have they assessed the impact of selecting from multiple simulations with high mean cross-correlation?

Minor

- The Abstract is missing a closing bracket. This should probably go after “alanine-serine-cysteine transporter”
- Please cite Isolde as it is related, perhaps along with references 9-14 in the introduction. Croll, Tristan Ian. 2018. “ISOLDE: A Physically Realistic Environment for Model Building into Low-Resolution Electron-Density Maps.” *Acta Crystallographica Section D: Structural Biology* 74 (6):519–30. doi: 10.1107/S2059798318002425.
- It should be mentioned at the end of the simulation methods section that the setup parameters are available on Zenodo, not just that they are available with a link that looks like another citation, just for easier readability.

Version 1:

Reviewer comments:

Reviewer #1

(Remarks to the Author)

The authors provided a comprehensive response to the concerns in the first review reports. They significantly improved the description and justification of their method, for example by additionally testing k-medoids with RMSD as a distance function and cryo_fit as an alternative fitting method, by marking the cluster representatives in the conformational space (fig 4B, 5B, 6B), or by expanding the methods section with useful information. While I am inclined to recommend the revised article for publication, two critical issues remain that, in my opinion, must be addressed before it can be accepted:

1. Robustness in selecting the final model: The authors addressed the concerns regarding the k-means clustering based on Cartesian coordinates by additionally testing k-medoids with RMSD as distance function and regarding the compound score by normalizing across all simulations. In my opinion, the modified versions should constitute the main workflow due to their stronger methodological foundation. However, the original version produces similar, or in some cases better, outcomes.

Nonetheless, it is important to evaluate whether the observed performance is robust and not attributable to chance:

1.1 How similar are the structures, which are closest to the cluster centroids, to the target structure in terms of RMSD? For example, consider evaluating the five closest structures for each cluster.

1.2 How similar are the best frames from the e.g. five MD-guided simulations with the highest mean cross-correlation? Evaluate the RMSD to the target structure.

This concern is motivated by very similar mean cross-correlations for a few simulations for all test cases, the results shown in Figure 5B and 6B as no cluster representatives are in the region of very high RMSD values to the known and target structure (maybe they are close to the cluster centroids), and by Figure S10 C: Although the structures indicated by the red and orange star are close in terms of combined GOAP score and cross-correlation, they significantly differ in their RMSD to the target structure. Namely, 8.26 angstrom compared to 3.65 angstrom. A table of RMSD values to the target structure is sufficient here.

2. In my initial review report, I requested a discussion on the computational runtime of the proposed method. I consider this an important aspect, as computational resources are valuable. Such information would help users assess whether the method is computationally feasible for their specific use cases. Please provide the computer architecture and rough runtimes for AlphaFold 2, the clustering process, and in particular the MD-guided simulations. There's no need for additional benchmarks; the runtimes from the analyses already conducted are sufficient.

Other minor issues are:

- I have not found an explanation of how the GOAP score and cross-correlation were normalized. Please add a description to the method section.
- The term 'normalized GOAP' is used for both, the normalization by sequence length and to limit the GOAP score to the range [0, 1]. Please differentiate more clearly.
- Figure 4B, 5B, and 6B: Highlight the cluster representative, which resulted in the final model after MD-guided simulations.
- Figure 2: Change 'centroid structures' to 'cluster representatives'.

Reviewer #2

(Remarks to the Author)

I believe the revised manuscript is significantly improved and I have no further concerns.

Reviewer #3

(Remarks to the Author)

In order to avoid further delay, I have reviewed the original comments of the third reviewer and the answers of the authors on behalf of the 3rd reviewer as well as the manuscript changes made.

In my opinion, the authors sufficiently addressed and implemented all comments of reviewer #3.

External Editor Kristyna Pluhackova

Erik Lindahl
Professor of Biophysics

Dear Editor,

Thank you for the thoughtful consideration of COMMSCHEM-25-0146. As outlined below, we believe we have responded to all reviewer concerns. In this response, editor and reviewer comments are gray (renumbered for clarity), and quotes from the revised manuscript are written in red.

REVIEWER #1

The paper by Tatiana Shugaeva et al. suggests an automated pipeline to obtain atomic protein coordinates for a given cryo-EM density map: Candidate models are generated with AlphaFold2 using subsampling of multiple sequence alignments and subsequently filtered via the GOAP score. Next, k-means clustering is used to sample 20 representative structures, which are then steered onto the target density with density-guided MD simulations. A final model is selected from these simulations based on cross-correlation and GOAP score. The suggested method seems very useful for proteins with multiple conformational states and is applicable to density maps with low to medium resolution. Moreover, the authors illustrate a clear improvement over a more standard approach (reference 14), where a single known atomic structure is steered onto the target density (alternative conformation). The proposed method is able to accurately model challenging local regions undergoing conformational changes. Overall, the idea of the method and the individual steps are clearly described and well structured. Additionally, the figures are well-designed (in particular Fig 3, 4, 5, 6, S3, and S4) and convey the most relevant information.

However, the following issues, listed in order of priority, must be resolved before publication:

1.1. k-means clustering is one of the most central points of the proposed method. However, it has multiple issues:

1.1.1. The candidate structures are initially aligned to the known structure. What if no known structure exists? This currently restricts the applicability of the method.

Response: We appreciate the enthusiastic assessment and constructive criticism. On this issue, we set out to address the specific and common use case of a protein whose structure is known in one state, but we want to resolve another one. However, we agree that the pipeline could also be applied to new proteins without known structures. Following the suggestion, we have added an additional approach clustering structures using *k*-medoids with RMSD as a metric between candidates, which does not require structural alignment to single known structure; please see the response to comment 1.1.2 for details.

1.1.2. MDAnalysis is used for k-means clustering of the candidate structures. Thus, the cartesian coordinates are flattened into a single feature vector, and subsequently, the KMeans class from sklearn with the Euclidean distance is used for the actual clustering. The large number of features (number of $C\alpha * 3$) may already be a problem. Moreover, the distances between candidates are not meaningful as they are based on the alignment to the known structure. I suggest using RMSD (or IDDT, GDT, QCS, ...) as a distance measure between candidates, then calculate the distance matrix, and directly use it for the KMedoids class (not KMeans as it does not allow for a distance matrix and relies on the Euclidean distance) of sklearn. This would also solve the problem of 1.1 as no alignment to the known structure would be necessary.

Response: We have added an alternative scheme using pairwise distance deviation and *k*-medoids, re-running the pipeline for all examples. To evaluate if this influences quality, we compared RMSD-to-target to those in our original protocol. In the best-correlated density-guided simulations using *k*-medoids, RMSD-to-target converged to similar values (new Fig. S2). Although this did not demonstrably improve quality, we describe both approaches, and have added relevant code to the Zenodo repository.

Results (p. 5):

To identify a limited set of plausible models representative of the generated ensemble, we then aligned the filtered models to the structure in the already-known state, and clustered them by the *k*-means method on the basis of Cartesian coordinates [20]. Finally, we used the model closest to each cluster centroid (cluster representative) for fitting to an experimental density, termed the target state.

Results (p. 8):

For cases where no structure in any alternative state is available, an alternative approach could be to cluster the initial AI-generated ensemble using *k*-medoids on the basis of pairwise internal distances [30]. In parallel trials of all three of our test cases using this technique (Fig. S2), RMSD from the target structure converged to similar values as with our primary approach, indicating this can be a useful alternative in relevant systems.

Methods (p. 20):

Erik Lindahl
Professor of Biophysics

To evaluate the extensibility of our approach to systems where no structure is known for any state, we also executed a parallel analysis of each AI-generated ensemble without initial alignment to any known state, directly computing a distance matrix representing RMSD in pairwise internal distances between all Ca atoms. We then used this matrix as input to the *k*-medoids clustering algorithm [30], which can be applied to arbitrary distance matrices, potentially more suitable for clustering on structural dissimilarity. This approach addresses potential limitations of our original approach using *k*-means clustering on aligned Cartesian coordinates, which could be overly biased by the known structure or otherwise fail to meaningfully capture structural similarity. In all three of our test cases (Fig. S2), RMSD from the target structure converged to similar values with or without known-state alignment, indicating that either approach can be successfully applied.

1.1.3. Candidates closest to the cluster centroids are selected. This approach is valid. However, Fig 4B, 5B, and 6B indicate that AlphaFold2 generated candidates with an RMSD less than 0.5, 1.3, and 2.0 angstroms to the target structures CLR, LAT1, and ASCT2, respectively. This is a lower RMSD than the best *k*-means candidates even after the density-guided MDs (which according to Fig 3 lead to further improvements over the AI-generated structures). From this, I conclude that *k*-means is not selecting optimal candidates (maybe due to points 1.1 and 1.2, and because *k*-means will not select the most diverse structures, but averaged structures for each cluster). Solving the max-min diversity problem might be an approach to selecting better candidates. Additionally, marking the selected 20 candidate structures in Fig 4B, 5B, and 6B would be helpful to evaluate the sampling method.

Response: We are grateful for this comment, as it facilitated our identification of an issue in our original RMSD calculations used in Figures 4B, 5B, and 6B. Specifically, we had previously used the “align” command in PyMOL, which can omit poorly aligned atoms from the calculation, underestimating the RMSD values. We have now re-run RMSD computation using the “rms_cur” command over a precise set of atoms specified for each system. Based on these, our initial AF2-generated ensembles included models with RMSD as low as 1.37 Å, 1.44 Å, or 2.35 Å with respect to target structures of CLR, LAT1, or ASCT2, respectively. In our fitting pipeline using *k*-means clustering with alignment to the known structure, the best-correlated density-guided MD simulations came from initial models with RMSD of 1.85 Å, 2.26 Å, or 4.43 Å, producing final models with RMSD of 1.25 Å, 1.44 Å, or 3.65 Å respectively. In our alternative pipeline using *k*-medoids clustering without structure alignment, the best-correlated density-guided MD simulations came from initial models with RMSD of 1.77 Å, 1.78 Å, or 3.42 Å, producing final models with RMSD of 1.46, 1.64, or 3.46 Å. Thus, either approach yielded models similar to or better than the best in the starting ensembles, in terms of (correctly calculated) RMSD-to-target structure. **Figures 4B, 5B, and 6B** have been updated with the recalculated values; all other figures and tables reporting RMSD values were generated using the correct method and remain unchanged.

It is indeed interesting that for ASCT2 the initial ensemble includes a model with modestly better RMSD than the final result. Whereas our clustering facilitated the selection of structurally diverse starting models for an accessible scale of density-guided simulations, in this case it also appears to divert attention from a good candidate. The challenge may have been underrepresented by our initial versions of **Figs. 4B, 5B, and 6B**, which cropped the ensemble of models to those within 4–12 Å RMSD from either known or target structures. We have now added an expanded version of each of these plots both in new **Fig. S11** and an inset in **Fig. S2** and each original panel, with markers for the cluster representatives used for density-guided simulations. These plots demonstrate more clearly that each ensemble includes models diverging up to 30 Å from one or both structures, making it challenging to identify a set of starting models that is sufficiently diverse to identify even a poorly sampled state, yet computationally tractable. Partly in response to Reviewer 2 below, we also add a new supplementary **Fig. S9** illustrating that models matching the target structure are not easily extracted from the initial filtered ensemble on the basis of cross-correlation alone. Although for ASCT2 an even-better model could in principle have been identified from the initial ensemble given unlimited time, compute and user resources, our approach nonetheless constitutes a consistently informative and accessible strategy in the anticipated experimental case where the target structure is unknown.

Results (p. 14):

Interestingly, the initial ensemble for ASCT2 included a model with a modestly better RMSD relative to the target structure (2.35 Å) than the final result from our *k*-means-based pipeline (3.65 Å) (Fig. 6), or from our alternative *k*-medoids clustering (3.46 Å, Fig. S2). As shown in previous sections, this issue did not appear with CLR or LAT1 where the best member of the starting ensemble included a model with 1.37 Å and 1.44 Å RMSD, respectively. After running our pipeline, our final models deviated equally or

Erik Lindahl
Professor of Biophysics

less from our *k*-means-based pipeline (1.25 Å and 1.44 Å, respectively), and from our alternative *k*-medoids clustering (1.46 Å and 1.64 Å, respectively) (Figs. S2, 4 and 5). Although in the case of ASCT2 our clustering method appeared to exclude a particularly optimal model, this hit could not have been identified given our presumed advance knowledge of only the target density, not the target structure.

Discussion (p. 16-17):

As illustrated by the wide range of structures generated in all test cases (Fig. S11), a critical challenge in our approach was to select a subset of starting models for density-guided simulations that would retain sufficient diversity to identify even a poorly sampled state, yet be tractable given finite time, compute and user resources. Although in one case (ASCT2) the initial ensemble included a model with a modestly better target-structure RMSD than the final result from our pipeline (Figs. S2 and 6), this hit could not have been identified in the anticipated experimental case where the target structure is unknown. Increasing the number of initial models or clusters could improve the likelihood of capturing states closer to the target, but at a steep computational cost. We suggest that future work should focus on enhancing AI sampling strategies to better explore biologically relevant conformational spaces, and improving clustering methods that balance diversity with proximity to functional states. 1.1.4. The authors correctly write 'and used the model closest to the cluster centroid for fitting' (not the centroid itself), but later in the article, the authors refer to the cluster representatives as the cluster centroids, which is not correct. This is also the case for Fig 2.

Response: We appreciate the reviewer pointing out the imprecise use of this term. We have now replaced "centroid" with "representative model" or "cluster representative" throughout the text and figure legends.

1.2. The density-guided simulation with the highest mean cross-correlation is selected and the frame with the highest compound score is chosen as the final model. Why is a selection of the best simulation necessary? Why not directly determine the frame with the highest compound score across all density-guided simulations? If the goal is to obtain a structure that optimally fits the cryo-EM density while maintaining high geometric and chemical quality, I believe this approach is more direct. The proposed method potentially excludes simulations that start with a low compound score, but progress to very high scores over the course of the simulation. Perhaps the authors are concerned about selecting unstable models. To address this, the first 500 or 1000 ps of the density-guided simulations could be excluded, or a more sophisticated analysis could be implemented.

Response: The calculation of compound score is an interesting concern. We normalize both GOAP score and cross-correlation to [0, 1] within simulations, largely as a straightforward way (short of introducing formal parameters) to avoid underweighting cross-correlation relative to GOAP score. As noted by the reviewer, an alternative would be to normalize both across all simulations of a given system, and select the best-scoring frame overall, rather than within the best-correlated trajectory. We have indeed tested this, but find that for CLR and LAT1, such an overall normalization selects precisely the same frame as our primary approach. For ASCT2, global normalization selects a model with modestly better GOAP score but worse cross-correlation, from a different trajectory than our main approach. Notably, RMSD-to-target structure was 8.26 Å for the final model using global normalization, vs. 3.65 Å normalizing within each trajectory, indicating that our approach yields an equivalent or better model in all our test cases. We have added a new **Fig. S10** showing globally normalized cross-correlations vs. GOAP scores in each system, highlighting the final model selected by both approaches.

Results (p 15):

Motivated by challenging characteristics particularly of the ASCT2 test case, we also evaluated alternative approaches to selecting a final model. Although GOAP is an established metric for model quality particularly for simulations in GROMACS, MolProbity is a common alternative in structural biology [42]. Monitoring MolProbity vs. GOAP scores during density-guided simulations resulted in final models with similar RMSDs relative to the target structure (3.39 Å vs. 3.65 Å respectively), indicating either metric could be effectively applied. Perhaps more critically, our primary approach relied on a compound score based on cross-correlation and GOAP scores normalized within each density-guided simulation; an alternative strategy would be to normalize each value across all simulations of a given system. For CLR and LAT1, such global normalization selected precisely the same frame as our primary approach (Fig. S10). For ASCT2, global normalization selected a model from a different trajectory than our primary approach, with a modestly better GOAP score but worse cross-correlation. Notably, RMSD with respect to the target structure was 8.26 Å for the final model using global normalization, vs. 3.65 Å when normalizing within each trajectory, indicating that our approach yields an equivalent or better model in all our test cases.

Erik Lindahl
Professor of Biophysics

1.3. The results suggest a clear improvement (RMSD) and slight improvements in model quality compared to a more standard approach (reference 14). However, it also requires significantly more runtime. Generating 1250 models via AlphaFold2, filtering based on the GOAP score, clustering, and 20 density-guided MD simulations compared to a single MD simulation with an equal amount of steps. I did not find this comparison in the article. While Fig 3 suggests that the standard method will not benefit from longer simulation times, maybe 20 simulations from the same starting structure with different starting velocities would provide better results? A discussion on this topic would be helpful.

Response: Good idea. We tried performing 20 density-guided simulations based on the same known structure of CLR — the case involving the most localized conformational change between known vs. target states, and thus requiring minimal improvement in overall RMSD. None of these simulations sampled frames with RMSD-to-target better than 2.3 Å. We illustrate this in new **Fig. S6**.

Results (p. 10):

In contrast, the best model from standard density-guided simulations of the inactive structure (known state) deviated 2.39 Å from the target globally, and by 6.42 Å locally, even over 20 independent replicates (Table 2, Fig. 3A, Fig. S6).

Discussion (p. 16):

In our test cases, standard density-guided MD simulations based on a single known structure produce models that deviate from the target structure by 2.33-10.16 Å, even when replicated up to 20 times for a single system.

1.4. According to Fig 3 the density-guided simulations were performed with different number of steps for the three test systems, namely 3000, 2000, and approximately 2400 for CLR, LAT1, and ASCT2, respectively. Were the 20 simulations within one test case performed with the same number of steps? What was used as a stopping criterion? A clearer description is necessary here.

Response: During density-guided simulations we use adaptive force scaling to ensure a steady increase in similarity between simulated and reference densities without manually tuning force-constant parameters. Simulations terminate automatically when forces on the system are too large for the integration time step, but at this point models are often distorted. To avoid this, we use a model-quality metric such as GOAP score to select an optimized frame (Yvonesdotter, *BpJ*, 2023).

Results (p. 5):

After rigid-body alignment to the relevant density, we subjected each representative model to density-guided MD simulations. Across each simulation, we monitored the cross-correlation to the target map as a model-fitting metric, and selected the simulation with the highest mean cross-correlation for further analysis. The implementation of adaptive force scaling for fitting in GROMACS means the run will continue until the overall forces of the system are too large to be compatible with the integration time step, which can introduce distortions. To instead identify a balanced frame, we also monitor the GOAP score as a quality metric for structural geometry [21], as reported in our previous study [22].

After normalizing both the fitting and geometry metrics to [0, 1], we subtracted the GOAP score from the cross-correlation to calculate a compound score for each simulation frame, with higher scores representing a combination of good fit (high correlation) and geometry (minimal penalty due to overfitting). We selected the frame with the highest compound score as a final model.

Methods (p. 21):

A scaling force factor of 10^3 kJ/mol was combined with adaptive force scaling in GROMACS, as the latter ensures a steady and automated increase in similarity between simulated and reference densities; simulations terminate automatically when overall forces on the system are too large to be compatible with the integration time step.

1.5. How were the parameters of the suggested automatized method selected? For instance, the number of initially generated models by AlphaFold2, the threshold for the GOAP score in filtering, or the number of clusters for k-means. If they were selected based on all three test cases, they no longer serve as true test cases since they were used for parameter optimization. Additional unseen test cases would then be necessary.

Response: Good point. We now elaborate on the relevance of assessing applicability to our test cases:

Discussion (p. 16):

Parameters for ensemble generation were selected based on previous work [46], while parameters such as GOAP-score cutoff and number of clusters were pre-determined on the basis of accessible resources.

Erik Lindahl
Professor of Biophysics

Alternative parameter values might enhance performance particularly in other systems, but were kept consistent in this work in order to assess applicability to our parallel test cases.

Methods (p. 17):

The MSA depth was subsampled according to previously proposed methods [19], using a single recycle and no energy minimization. As optimized in previous work [46], we used MSA depths in the range of 5-30 to generate 50 models at each depth, resulting in 1250 initial models for each system.

Model clustering and rigid alignment

We discarded models with normalized scores above -100, a round threshold value selected arbitrarily, but which proved to filter models with the worst geometry in all three test cases (Fig. S4). The remaining models were aligned to the structure assigned as the already-known state using PyMOL [67], and clustered using the *k*-means method on the basis of C α distances in the MDAnalysis Python package [20]. Evaluation of 20 cluster representatives for each system was determined in advance to be accessible given the available compute resources; this number evidently enabled sufficient structural diversity to improve fitting in all our test cases, within practical limits of reproducibility.

1.6. Minor issues and suggestions:

1.6.1. Filtering with the GOAP scores is not shown in Fig 2.

Response: We have added a step “Model filtering” in **Figure 2**.

1.6.2. Fig 1 lacks a precise description: 1. Differentiation between top and bottom is missing. 2. Descriptions of the arrows pointing to the yellow regions are missing. 3. Bottom: The density is sometimes smaller than the atomic protein. 4. Bottom: Labelling of the arrows - for example with 'Steered MD' - is missing. 5. The authors write 'based on a previously known structure (ribbons) may be challenging for a given protein (blue)'. Ribbons and blue both refer to the same protein? Also, everything (proteins + densities) is blue in the figure.

Response: We have updated the legend to **Figure 1** as suggested.

Fig. 1 (p. 4):

Top, general representation of the common experimental problem in which a target cryo-EM density (center) is resolved for a protein with a known structure (left) that differs from the target in one or more regions, indicated by arrows; an AI-generated ensemble (right) may include models that more closely approximate the target. *Bottom*, detailed representation of a case in which density-guided simulations, represented by forward arrows, fail to fit a plausible model based on the known structure (left), but succeed on the basis of at least one member of the AI-generated ensemble (right).

1.6.3. Fig 2 has rectangles around the protein emojis, which can be easily mistaken for the density. Thus, I suggest to remove all rectangles from the figure.

Response: Most rectangles around protein cartoons have been removed. The ones that were left have been marked in a lighter color to avoid confusion.

1.6.4. Fig 3 depicts two structures for A, B, and C. Without reading the text, it is unclear what the structures represent. Adding text like 'known structure' and 'final model' to the figure would be helpful.

Response: “Known structure” and “final model” labels have been added to each panel of **Figure 3**.

1.6.5. Fig 4, 5, and 6 are great. They visualize the key results exceptionally well!

Response: Appreciated!

1.6.6. Table 2 provides an excellent overview of the evaluation criteria and results. However, the corresponding description in the main text is somewhat imprecise. For example, the authors write 'our generated ensemble was within 3.2 angstroms C α RMSD from the outward-open target structure, while the best model based on fitting the inward-open known structure deviated more than 10.1 angstroms' to describe the deviations of 3.14 angstroms and 10.16 angstroms. I suggest using the exact values to match Table 2 and avoid unnecessary comparisons with it.

Response: The **Results** text has been updated to precisely match the values in Table 2.

Results (p. 9):

We then compared our approach to standard flexible fitting starting from the known structure of CLR in an inactivated state. The final model generated in our approach deviated from the activated experimental structure (target state) by only 1.25 Å C α RMSD (Table 2, Fig. 3A), and included the active-state kink

Erik Lindahl
Professor of Biophysics

(Fig. 4G), such that even local deviation in TM6 was limited to 1.6 Å (Fig. S5). In contrast, the best model from standard density-guided simulations of the inactive structure (known state) deviated 2.39 Å from the target globally, and by 6.42 Å locally, even over 20 independent replicates (Table 2, Fig. 3A, Fig. S6). Similarly, another flexible fitting method not based on GROMACS, `cryo_fit` [34], also did not result in good agreement between the fitted model and the target structure (global RMSD from the target of 3.27 Å after fitting, Table S1). Compared to standard known-state fitting, our approach also produced moderately better model quality (-95.9 vs. -91.2 normalized GOAP, 0.23 vs. 1.36 Clash, 1.41 vs. 1.88 MolProbity scores) and fit to the target density (0.79 vs. 0.75 cross-correlation) (Table 2).

Results (p. 10):

Conversely, TM6 remained largely straight after density-guided simulations of the inactive known structure (Fig. 4F), such that local deviation from the target structure in this region was 6.42 Å (Table 2 and Fig. S5).

Results (p. 11):

The final model based on our generated ensemble was within 1.44 Å Cα RMSD from the inhibited target structure (PDB ID 7DSQ [28]), while the best model based on fitting the inward-open known structure deviated by 2.33 Å for GROMACS density fitting (Table 2, Fig. 3B) or 2.96 Å for `cryo_fit` tool [34]. Our pipeline model also deviated by only 1.52 Å in the TM1/TM6 region, and was compatible with binding diiodo-Tyr, despite the absence of this inhibitor model and its density during fitting (Table 2, Fig. S8). Conversely, the known-state fitted model deviated over 4.08 Å (Table 2) in the TM1/TM6 region, and oriented a Tyr residue to clash with the anticipated inhibitor pose. Compared to standard known-state fitting, our generative ensemble setup produced modestly better model quality (-103.6 vs. -100.2 normalized GOAP, 1.41 vs. 1.98 Clash, 1.34 vs. 2.01 MolProbity scores) and fit to the target density (0.59 vs. 0.52 cross-correlation).

Results (p. 13):

The final model based on our generated ensemble was within 3.65 Å Cα RMSD from the outward-open target structure, while the best model based on fitting the inward-open known structure deviated by 10.16 Å for GROMACS density fitting or 11.73 Å for `cryo_fit` tool (Tables S1 and 2). The overall RMSD value converged to similar values with or without blurring the original cryo-EM maps during our simulation in the ensemble approach Fig. S1.

As in previous cases, our approach also produced modestly better model quality (-99.8 vs. -95.8 normalized GOAP, 0.62 vs. 1.23 Clash, 1.06 vs. 1.63 MolProbity scores) and fit to the target density (0.63 vs. 0.57 cross-correlation).

1.6.7. It is initially unclear which clustering algorithm was used. This should be mentioned directly in the abstract and at the beginning of the results section under 'We then clustered these to identify...!'

Response: We now explicitly describe the clustering algorithm in the **Abstract** and **Results**.

Abstract (p. 1):

Here, we introduce a refinement approach in which i) several initial models are generated by stochastic subsampling of the multiple sequence alignment (MSA) space in AlphaFold2, ii) the resulting models are subjected to structure-based k-means clustering, iii) density-guided molecular dynamics simulations are performed from the cluster representatives, and iv) a final model is selected on the basis of both map fit and model quality.

Results (p. 5):

To identify a limited set of plausible models representative of the generated ensemble, we then aligned the filtered models to the structure in the already-known state, and clustered them by the k-means method on the basis of Cartesian coordinates [20].

REVIEWER #2

Shugaeva et al. present a refinement approach for modeling atomic coordinates into cryo-EM maps, combining generative AI models with density-guided molecular dynamics simulations. While the study demonstrates improved fitting accuracy for three membrane proteins undergoing conformational transitions, several aspects warrant critical consideration regarding novelty, methodology, and scope.

2.1. The authors state in the abstract that their method combines "generative AI" with simulation-based refinement to address challenges in building alternative states of membrane proteins. However, the generative component relies entirely on AlphaFold2, a pre-existing software not developed by the authors. As such, referring to it as "Modeling cryo-EM structures in alternative states with AlphaFold2 based models and density-guided simulations" would be more accurate. Although, the stochastic subsampling of MSA

Erik Lindahl
Professor of Biophysics

depth within AlphaFold2, as described in the Results section ("we first used stochastic subsampling of input MSA depth in AlphaFold2...") is admittedly a novel approach. Similar services, such as DeepMainMast (Terashi et al., *Nature Methods* 2024), already offer cryo-EM model refinement using AlphaFold2 structures and density information. It should be more explicitly clarified in the manuscript what distinguishes the authors' implementation from these existing services.

Response: Following the reviewer's suggestion, we have now changed the title of the manuscript to "Modeling cryo-EM structures in alternative states with AlphaFold2 and density-guided simulations." To better distinguish our approach from existing methods, we have also modified our **Discussion**.

Discussion (p 18):

Some *de-novo* methods of cryo-EM structure determination, such as those implemented by DeepMainMast [47] and similar tools [48-58], also incorporate prediction tools like AlphaFold or RoseTTAFold to generate protein or fragment models. However, even with AI-generated models, predicting divergent or rare protein conformations may be challenging [59]. A distinctive feature of the approach in this work is the clustering of a substantial and diverse ensemble of full-length models into an accessible subset of broadly distributed models; only one such cluster representative needs to be a good approximation of the target structure to enable accurate refinement. Thus, our approach might be particularly applicable to proteins with multiple metastable states or otherwise flexible features.

2.2. The rationale for structure-based clustering after generating AlphaFold2 models is not fully clear to me. If the goal is to identify structures closer to the target map, it seems logical to directly initiate MD simulations from the AlphaFold2 structure exhibiting the highest initial correlation with the target density. The added complexity of clustering and selecting centroid structures requires some further explanation.

Response: Indeed, in our initial explorations of AlphaFold2-based structure modeling we sought to identify accurate models from an entire filtered ensemble on the basis of cross-correlation alone. However, the best-correlated models proved to deviate substantially from the target structure. We now illustrate this point in newly added **Figure S9**, and explain it in **Results**.

Results (p. 14):

Alternatively, we attempted to select models approximating the target structure directly from the filtered ensemble, prior to clustering or simulations, on the basis of cross-correlation to the target density; however, the best-correlated models for ASCT2 were visibly different, and exhibited relatively high RMSD (9.57-11.39 Å), with respect to the target structure (Fig. S9). Thus, knowledge of the target density was insufficient to identify plausible models prior to clustering and simulations, supporting the overall utility of our approach.

2.3. I also believe the choice of the Generalized Orientation-Dependent All-Atom Potential (GOAP) score as a structure geometry metric requires further explanation. The authors should provide a comparative analysis of GOAP against other established metrics like MolProbity, Q-scores, Ramachandran outliers, or clash scores. A discussion of GOAP's strengths and weaknesses, as well as its sensitivity to specific structural features, would enhance the validity of the results.

Response: Although GOAP score is a straightforward metric established for monitoring density-guided simulations in GROMACS, we appreciate the suggestion of comparing alternative quality measures. To this end, as also described in response to comment 1.2, we now report in **Results** the comparative selection of a final model for ASCT2 on the basis of MolProbity rather than GOAP score with effectively equivalent RMSD to the target structure, and contextualize the metric choice in **Discussion**.

Results (p. 15):

Although GOAP is an established metric for model quality particularly for simulations in GROMACS, MolProbity is a common alternative in structural biology [42]. Monitoring MolProbity vs. GOAP scores during density-guided simulations resulted in final models with similar RMSDs relative to the target structure (3.39 Å vs. 3.65 Å respectively), indicating either metric could be effectively applied.

Discussion (p 17):

Similarly, we adopted GOAP score as an assessment of geometric quality based on previous work [22] and straightforward implementation in GROMACS density-guided simulations. However, GOAP scoring is based on a reference set of >1000 divergent structures resolved to <2 Å [21], and may be accordingly biased towards characteristics of soluble proteins determined by X-ray crystallography; in some cases, alternative metrics such as MolProbity score [42] may be preferred.

2.4. In the discussion, the authors mention their method's applicability "among other things to multi-state proteins" (page 13). This statement is somewhat vague and should be clarified with specific examples.

Erik Lindahl
Professor of Biophysics

Additionally, the limitations of the method, ex., concerning nucleic acids, nucleic acid-protein complexes, or systems with limited structural data, should be more explicitly stated.

Response: Following the reviewer's suggestion, we have now added to **Discussion** more specific examples where our approach might be useful, as well as possible limitations.

Discussion (p. 17):

The success of our approach in these systems indicates its applicability to multiple dynamic proteins, including physiologically critical membrane transporters and receptors. It is likely transferrable to a variety of proteins that undergo conformational transitions, though current limitations in AlphaFold2 restrict its use in more complex biological assemblies, for example involving nucleic acids; fortunately, AlphaFold3 [37] has shown potential for modeling such systems.

2.5. The authors assert that "flexible fitting may fail to accurately refine a complex target for which the initial model substantially differs" (page 13). However, they do not provide convincing evidence that their method outperforms established flexible fitting methods, in particular those specifically designed for such scenarios. A comparison against benchmarked methods like the one described in reference 12 of the manuscript (Igaev et al., *eLife* 2019), possibly using the same benchmarks found in that reference, would greatly help to clarify the issue.

Response: We appreciate the admonition that the original statement overly generalized the comparisons. We have now refined the **Discussion** to focus on comparison to standard density-guided simulations based on a known structure. We also acknowledge more explicitly some alternative approaches to flexible fitting documented in the literature, including the suggested reference [12], some of which could also be applicable to our experimental scenario; future development of user interfaces will likely offer more rigorous benchmarking of methods, but was respectfully deemed beyond the scope of this work.

Discussion (p. 16):

In this study, we propose and implement an approach that leverages AI generation of protein models along with density-guided MD simulations for atomistic modeling into cryo-EM maps. Automated structure determination can in some cases be achieved by rigid or flexible fitting of known structures (refs 9-14). However, each of our test systems illustrates a challenging case where standard flexible fitting in GROMACS or alternative cryo_fit implementation based on a known structure fails to accurately refine a complex target in a substantially different state.

Discussion (p. 16):

In cases for which restricted homology or intrinsic disorder prevents approximation of the target conformation [44], multi-step approaches using progressive resolution [45], gradual parameter changes [12] or enhanced sampling [10, 11] may prove valuable.

2.6. The validation strategy relies on comparing the generated models to experimentally determined structures of the target state. However, I see it, the method essentially uses AlphaFold2 to identify a state closer to the target, followed by standard density-guided MD simulations. Any method capable of generating a similar starting point would likely achieve comparable results. The manuscript should address this by demonstrating that the method outperforms alternative approaches for generating suitable starting models, such as automated fitting procedures in software packages like Phenix, Isolde, or Rosetta.

Response: We strongly agree there are many good fitting methods, and that given a starting model sufficiently close to the target, all of them (including density-guided simulations) should perform a fit in an automated, accurate fashion - but the good starting models are sometimes unavailable, which is where a good-quality predicted ensemble becomes useful. Indeed, the challenge in our test cases was to identify close-to-target structures when least a portion of the protein was difficult to predict in the intended state. To directly compare the ensemble approach to *de-novo* model generation, we now document in new **Fig. S3** and **Table S1** predictions for CLR, LAT1 and ASCT2 using the state-of-the-art tool ModelAngelo [8]. Since they are generated on the basis of the target densities, the resulting predictions are indeed partly similar to the target structures; however, they include only a subset of each protein. For CLR, 85% of the protein is predicted, leaving unbuilt one end of the key kinked TM6 helix. For LAT1 and ASCT2, predictions cover only 43% and 64% of the protein respectively, excluding entire helices and portions of key transitional regions. This is in contrast to the final models from our approach, which include the entire input amino-acid sequence including dynamic regions. Thus, at least in context of this tool and test systems, *de-novo* prediction would for now require substantial subsequent user input to complete model building and fitting. We cover these comparisons in the fig/table and have the updated text:

Results (p. 7):

Erik Lindahl
Professor of Biophysics

On the other hand, AI-based *de-novo* prediction without AlphaFold2 was challenging in all test cases: predictions using the state-of-the-art tool ModelAngelo [8] included only a subset (43%-85%) of each protein, leaving portions of key transitional regions unbuilt (Fig. S3). Substantial subsequent user input would be required to complete model building and fitting, in contrast to the final models from our approach, which covered the entire input sequences. Accordingly, we focused the remainder of our study on documenting our approach based on AI-driven generation of a substantial and diverse ensemble, followed by clustering and screening of density-guided simulations to identify a representative models capable of fitting to the target state.

Methods (p. 19):

To assess the applicability of *de-novo* model generation in these systems, we also attempted to predict structures based on their target densities in ModelAngelo [8]. We used the default ModelAngelo build command for the case where both the cryo-EM map and the protein sequence were available, following the instructions provided in the ModelAngelo GitHub repository (<https://github.com/3dem/model-angelo>).

2.7. Overall, I feel the manuscript lacks a clear focus regarding its intended application. Is the method primarily aimed at flexible fitting, geometry optimization, modeling distant states, or ensemble refinement? The data presented and the validation strategy are insufficient to support its effectiveness across all these applications. To position the work more clearly, the authors should clarify in the introduction which specific problems they are trying to address and then provide corresponding benchmarks against state-of-the-art methods for those problems throughout the manuscript.

While the presented approach shows promise for modeling cryo-EM structures in alternative states, the manuscript requires further work to address concerns regarding novelty, methodology, validation, and scope. A more rigorous comparison to existing methods, a clearer articulation of the method's strengths and limitations, and a more focused validation strategy are essential to establish the true value of this work.

Response: As similar concerns were raised by other reviewers and in other specific comments from this one, we have now invested in clarifying the purpose of our study in Introduction and Discussion. As documented in response to **comments 1.1.2, 1.2, 1.3, 2.2, 2.3, 2.6, 3.2 and 3.3**, we have added comparative Results using *de-novo* model building or alternative methods of starting-model selection, clustering, density blurring, flexible fitting, geometry scoring, metric normalization, and benchmark replication. As described in response to **comment 2.5**, we also revise some initial comparisons to more specifically define the scope of our study. Similarly, we now state early in the **Introduction** that the critical benchmark for our approach is density-guided simulations starting from a known structure in GROMACS, as subsequently documented in Figures 4, 5, 6, and Table 2.

Introduction (p. 3):

Taking advantage of recent developments in neural network-based structure prediction [18], we use stochastic subsampling of the multiple sequence alignment (MSA) depth in AlphaFold2 [19] to generate a broader ensemble of potential starting structures, then run density-guided MD simulations from cluster centroids to select an optimized structure for a given experimental density. We demonstrate the applicability of this approach to three test cases, all membrane proteins that undergo conformational changes between experimentally resolved states. As reference, default density-guided simulations in GROMACS failed to accurately fit the structure built in one state to the density resolved in another. In contrast, using our new approach, we successfully resolve state-dependent differences including the bending of an individual helix, rearrangement of neighboring helices, and reformation of an entire domain, demonstrating applicability to a range of systems and conformational dynamics.

REVIEWER #3

Shugaeva et al. present a novel approach for fitting atomic models to cryo-EM maps based on a recently established approach for generating models in an ensemble of multiple conformations with AlphaFold2 combined with density-guided simulations. The paper is well-written and shows that this approach seems very promising in cases where atomic modelling is challenging as demonstrated for 3 case studies from the EMDB. However, the approach seems to be rather manual at this point (with several independent steps including a manual analysis one for selecting fitting trajectories and best frames) and would greatly benefit from automation to make it reliably usable. There is some mention right at the end of the methods that "Python 3.9 scripts were utilised to combine pipeline steps, conduct analyses and generate figures" but it would be great if these were described throughout and made available with instructions. Some of these seem to be available in the Zenodo record but this is not completely clear. Providing a complete user-

Erik Lindahl
Professor of Biophysics

friendly implementation would make this study much more valuable to the general structural biology community. I also have several more specific comments.

Major

3.1. The current manuscript does not discuss the difficulty of using this approach for non-expert users, which seems to be considerable. There is a mention in the discussion that the authors implement an approach but otherwise we only see brief descriptions of the steps outside of the methods section, which seems rather involved with several packages that need to be installed and used.

Response: We appreciate the reviewers generally enthusiastic assessment! In reference to the more specific concern, we have now revised language in the **Discussion** to clarify the manual nature of some steps, and revised the **Methods** and **Availability** to identify specific scripts used in model generation, clustering and density-guided simulations.

Discussion (p 17):

Python scripts are available for several key steps in our approach, which could likely be further automated in future. Manual tools such as Phenix or Coot might further improve model quality, though we deliberately avoided such additional steps in order to optimize our approach for cases of limited structural biology or system-specific expertise.

Methods (p 19-20):

Python scripts for model clustering and rigid alignment are available on Zenodo (<https://doi.org/10.5281/zenodo.14749349>).

All simulation setup parameters and preparation scripts have been made available on Zenodo (<https://doi.org/10.5281/zenodo.14749349>). Density fitting tutorial in GROMACS is available at <https://tutorials.gromacs.org/density-fit-simulation.html>.

Data and Materials Availability (p 22):

All data needed to evaluate the conclusions are present in the paper or the Supplementary Information. The generated models with density-guided simulation input files and trajectories can be found on Zenodo: 14749350 (<https://zenodo.org/records/14749350>). A tutorial for the density-guided simulations performed in GROMACS is available at <https://tutorials.gromacs.org/density-fit-simulation.html>.

3.2. The authors mention in the end of the first section of the results that they applied a Gaussian blur to the target densities to mimic lower resolution situations. However, there is no mention of the impact of this blurring in the case study results. Please include an additional supplementary figure for each case study comparing the results with and without the blur. It can often be the case that density-guided simulations struggle in higher resolution cases due to the more rugged optimization landscape. It may well be that this is not the case here as the best models from the ensemble are already very close to the target and this would be good to show. On the other hand, if the authors believe that this Gaussian blur is useful for better fitting then this should be included as part of the approach. There is an indication of this in the discussion.

Response: We have now performed new simulations using the unblurred original maps for all the three test cases. As shown in newly added **Figure S1**, fitting to densities with or without blurring results in final models with similar RMSD values relative to the target structure. This demonstrates that blurring is not necessary, but also indicates the approach can be applied whether or not high-resolution cryo-EM maps are available. In addition to the new figure, we now emphasize this in **Results** and **Discussion**.

Results (p. 7):

We tested fitting to densities with or without a 1 Å Gaussian blur applied, and observed comparable convergence to target structures (Fig. S1). To emphasize the applicability of our approach to medium-resolution maps, we focused subsequent analyses on fitting to the blurred densities.

Discussion (p. 19):

Target densities tested in this work were resolved to 2.3–3.4 Å overall, and were subjected to an additional 1 Å Gaussian blur prior to fitting. This blurring did not affect the final model quality; simulations using either original or blurred maps converged to similar RMSD values relative to the target structures (Fig. S1), consistent with indications that modest differences in map resolution have limited impact in context of a well matched initial model. This observation also supports the applicability of our approach even when only moderate resolution maps are available.

Methods (p. 21):

After merging selected regions, to explore the applicability of our approach to medium-resolution maps, we added a 1 Å Gaussian blur to each target density; however, we noted that RMSDs converged to similar values with or without blurring (Fig. S1).

Erik Lindahl
Professor of Biophysics

3.3. While this approach outperforms the standard density-guided simulation approach with a single model that is far from the target density when using their all-atom density-guided simulation implementation in Gromacs, no comparison is made to any other method, such as MDFF, normal mode-based flexible fitting, Rosetta or traditional real space refinement with Phenix.

Response: This is a fair point to check. We have now performed a new set of simulation with another flexible fitting method called `cryo_fit`, for all our test cases. As highlighted in **Table S1**, `cryo_fit` also failed to correctly fit the structural model to the experimental map when starting from an experimental structural from the known state. We cover this with new text in the **Results**, and **Methods**.

Results (p. 9):

Similarly, another flexible fitting method not based on GROMACS, `cryo_fit`[34], also did not result in good agreement between the fitted model and the target structure (global RMSD from the target of 3.27 Å after fitting, Table S1).

Results (p. 11):

The final model based on our generated ensemble was within 1.44 Å C α RMSD from the inhibited target structure (PDB ID 7DSQ [26]), while the best model based on fitting the inward-open known structure deviated more than 2.3 Å for GROMACS density fitting (Table 2, Fig. 3B) or 2.96 Å for `cryo_fit` tool [34].

Results (p. 1):

The final model based on our generated ensemble was within 3.2 Å C α RMSD from the outward-open target structure, while the best model based on fitting the inward-open known structure deviated more than 10.16 Å for GROMACS density fitting (Table 2) or 11.73 Å for `cryo_fit` tool (Table S1 and 2).

Methods (p 21):

To assess the performance of alternative flexible fitting methods on our known experimental states, we used `cryo_fit`[34], an automated flexible fitting tool integrated into the Phenix software suite[76]. All default options were used, except the “Multiply EM weight by this number” field, which was set to 100 to ensure that the fitting force was sufficiently strong to balance the MD force field and stereochemistry preserving constraints.

3.4. The difficulty of obtaining the target models in the PDB that were used for comparison and the need for the approach should also be discussed in comparison to the proposed approach. In particular, the CLR system required Rosetta and MDFF as well as several rounds of manual fitting with Coot and real space refinement with Phenix.

Response: We have now additional text reflecting the reviewer’s point in the **Discussion**.

Discussion (p. 18):

Within the scope of presently accessible tools, we expect our work to represent a particularly relevant approach in cases of limited structural-biology or system-specific expertise; for comparison, the target structure of CLR was determined using Rosetta and MDFF plus several rounds of manual fitting in Coot and real-space refinement using Phenix [24].

3.5. The 2nd and 3rd case studies still show fairly low cross-correlations, and the 3rd case still has an RMSD worse than 3 Å, even with the proposed approach. Have the authors tried refining the output models further with standard approaches and is this successful in improving them? This would be important to confirm that the solutions are not stuck in local minima.

Response: We appreciate this practical insight, and agree that additional refinement could further improve the final model. However, we avoided such steps to keep our approach accessible to non-structural biologists. We have added new text in the **Discussion** reflecting this point.

Discussion (p. 17):

Manual tools such as Phenix or Coot might further improve model quality, though we deliberately avoided such additional steps in order to optimize our approach for cases of limited structural biology or system-specific expertise.

3.6. The authors always pick the simulation with the highest mean cross-correlation but in many cases several simulations have similar mean cross-correlation. Have they assessed the impact of selecting from multiple simulations with high mean cross-correlation?

Response: This is an important point, raised also by reviewer 1. Please see our response to **comment 1.2** for a detailed reply, including new **Fig. S10** and text in the Result section (p 15).

Stockholm
University

Erik Lindahl
Professor of Biophysics

3.7. Minor

3.7.1. The Abstract is missing a closing bracket. This should probably go after “alanine-serine-cysteine transporter”

Response: Fixed.

3.7.2. Please cite Isolde as it is related, perhaps along with references 9-14 in the introduction. Croll, Tristan Ian. 2018. “ISOLDE: A Physically Realistic Environment for Model Building into Low-Resolution Electron-Density Maps.” Acta Crystallographica Section D: Structural Biology 74 (6):519–30. doi: 10.1107/S2059798318002425.

Response: Good point; we’ve added a citation in the suggested place.

3.7.3. It should be mentioned at the end of the simulation methods section that the setup parameters are available on Zenodo, not just that they are available with a link that looks like another citation, just for easier readability.

Response: We now explicitly cite the Zenodo link in this section of Methods.

Once again, thank you for very constructive feedback.

Sincerely,

Erik Lindahl

Thank you for the thoughtful consideration of our manuscript (COMMSCHEM-25-0146). As demonstrated in the attached revision and outlined below, we believe we have responded to the listed concerns of reviewer 1, including edits to the **Results, Methods, Figure 2, 4B, 5B, and 6B, Supplementary tables 1 and 2**. In this response, reviewer comments are written in gray (renumbered for clarity), and quotes from the revised manuscript are **red**.

The authors provided a comprehensive response to the concerns in the first review reports. They significantly improved the description and justification of their method, for example by additionally testing *k*-medoids with RMSD as a distance function and *cryo_fit* as an alternative fitting method, by marking the cluster representatives in the conformational space (fig 4B, 5B, 6B), or by expanding the methods section with useful information. While I am inclined to recommend the revised article for publication, two critical issues remain that, in my opinion, must be addressed before it can be accepted:

1. Robustness in selecting the final model: The authors addressed the concerns regarding the *k*-means clustering based on Cartesian coordinates by additionally testing *k*-medoids with RMSD as distance function and regarding the compound score by normalizing across all simulations. In my opinion, the modified versions should constitute the main workflow due to their stronger methodological foundation. However, the original version produces similar, or in some cases better, outcomes. Nonetheless, it is important to evaluate whether the observed performance is robust and not attributable to chance:

1.1 How similar are the structures, which are closest to the cluster centroids, to the target structure in terms of RMSD? For example, consider evaluating the five closest structures for each cluster.

Response: With thanks for this suggestion, we have added to our Zenodo deposition (<https://doi.org/10.5281/zenodo.14749349>) tables of RMSDs versus target for the five structures closest to the centroid of each cluster in each test system, using either *k*-means or *k*-medoids approaches. For ease of readability, we have also added a new **Table S1** to the manuscript, reporting these RMSDs for each best-performing cluster. As seen in these tables, the structures surrounding each centroid appear to be convergent, such that stochastic differences leading to selection of a modestly different starting structure are not expected to dramatically influence the final result using either clustering approach. We now highlight this observation in **Results** (p. 7):

Both *k*-means and *k*-medoids clustering appeared robust in enabling selection of starting models for density-guided simulations that converge to low RMSD-to-target: in the cluster producing the final best frame for each test system, the five structures closest to the centroid exhibited a spread in RMSD-to-target of ≤ 0.5 Å (Table S1).

1.2 How similar are the best frames from the e.g., five MD-guided simulations with the highest mean cross-correlation? Evaluate the RMSD to the target structure.

This concern is motivated by very similar mean cross-correlations for a few simulations for all test cases, the results shown in Figure 5B and 6B as no cluster representatives are in the region of very high RMSD values to the known and target structure (maybe they are close to

the cluster centroids), and by Figure S10 C: Although the structures indicated by the red and orange star are close in terms of combined GOAP score and cross-correlation, they significantly differ in their RMSD to the target structure. Namely, 8.26 angstrom compared to 3.65 angstrom. A table of RMSD values to the target structure is sufficient here.

Response: Following the reviewer's suggestion, we have now added **Table S2**, showing best-frame RMSDs versus the target for the top-5 simulations ranked by mean cross-correlation. As seen in the table, the best frames from different density-guided simulations diverged substantially from one another. Moreover, the simulation with the highest mean cross-correlation produced the best-scoring frame with the lowest target RMSD (or in one case, within 0.15 Å of the lowest) among the five best-correlated simulations, indicating our clustering approach was effective in distinguishing successful versus unsuccessful starting models. We now specify this point in **Results** (p. 7):

Both clustering approaches were also effective in seeding distinct conformations. Among the five best-correlated density-guided simulations in each system, the frames with highest compound score sample different parts of structure space: the three different systems exhibit spreads in RMSD-to-target from 1.0 Å to 6.5 Å (Table S2). In all cases, when selecting the highest-scoring frame from the simulation with the best mean cross-correlation, it produced the lowest RMSD-to-target (or within 0.15 Å from the lowest in one case), indicating that both clustering approaches were effective in distinguishing starting models capable of converging to the target.

2. In my initial review report, I requested a discussion on the computational runtime of the proposed method. I consider this an important aspect, as computational resources are valuable. Such information would help users assess whether the method is computationally feasible for their specific use cases. Please provide the computer architecture and rough runtimes for AlphaFold 2, the clustering process, and in particular the MD-guided simulations. There's no need for additional benchmarks; the runtimes from the analyses already conducted are sufficient.

Response: We now report runtimes for all computationally intensive steps in **Methods**, as detailed below.

Ensemble generation (p. 19):

AlphaFold2 inference was executed on an NVIDIA DGX A100 GPUs, requiring ~40 hours per test case.

Model clustering and rigid alignment (p. 20):

Clustering ran on a local cluster using one x86 CPU node within minutes.

Density-guided simulations (p. 20):

Each simulation ran on a local cluster using two x86 CPU cores and one NVIDIA GeForce RTX 2080 GPU. For each test case, we ran 20 systems in parallel with a total wall-clock time of 10–20 hours.

3. Other minor issues are:

3.1 I have not found an explanation of how the GOAP score and cross-correlation were normalized. Please add a description to the method section.

Response: We have now added a more detailed description of GOAP score and cross-correlation normalization to **Methods** (p. 21):

The compound score was defined as a combination of the GOAP score and the cross-correlation, enabling optimization of protein geometry as well as map fit. First, both the cross-correlation and the GOAP score absolute value were subjected to min–max normalization over the interval [0, 1]. The final compound score was then calculated as the sum of the normalized GOAP score and the normalized cross-correlation.

3.2 The term 'normalized GOAP' is used for both, the normalization by sequence length and to limit the GOAP score to the range [0, 1]. Please differentiate more clearly.

Response: We are grateful for the opportunity to clarify this terminology. We have now updated captions in **Figures 2, 4, 5, 6** and descriptions in **Results** and **Methods**, so that “scaled GOAP score” consistently refers to the initial value calibrated to the amino-acid sequence length, while “normalized GOAP score” refers to further normalization in the [0, 1] range.

3.3 Figure 4B, 5B, and 6B: Highlight the cluster representative, which resulted in the final model after MD-guided simulations.

Response: The representatives have been highlighted as orange triangles, and captions to **Figures 4, 5 and 6** updated (p. 9, 12, 15):

The orange triangle marks the representative that achieved the best fit.

3.4 Figure 2: Change 'centroid structures' to 'cluster representatives'.

Response: We have now implemented the suggested edit in **Figure 2**.